# Are We on the Right Way for Assessing LLM-as-a-Judge?

## Abstract

LLM-as-a-Judge has been widely adopted as an evaluation method and served as supervised rewards in model training. However, existing benchmarks for LLM-as-a-Judge are mainly relying on human-annotated ground truth, which introduces human bias that undermines the assessment of reliability and imposes scalability constraints. To overcome these limitations, we introduce SAGE, a novel evaluation suite that assesses the quality of LLM judges without necessitating *any* human annotation. Inspired by axioms of rational choice theory, SAGE introduces two new lenses for measuring LLM-as-a-Judge: local self-consistency (pair-wise preference stability) and global logical consistency (transitivity across a full set of preferences). We curate a dataset of 650 questions by combining structured benchmark problems with real-world user queries. Our experiments demonstrate both the intrinsic stability of our metrics and their high correlation with supervised benchmarks like LLMBar and RewardBench2, confirming SAGE's reliability as an evaluation suite for the robustness and accuracy of LLM-as-a-Judge. Based on SAGE, we reveal that current *state-of-the-art* LLMs exhibit significant robustness deficiencies when acting as judges; even the top-performing models, Gemini-2.5-Pro and GPT-5, fail to maintain consistent preferences in nearly a quarter of difficult cases. We attribute this to a new phenomenon called **situational preference** which explains why explicit rubrics or criteria can help model judge consistently across answer pairs. Our further analysis shows that fine-tuning LLM-as-a-Judge is an unreliable method which further induces biases, while multi-agent judges, deep reasoning can enhance performance through different means.

## 1 Introduction

The LLM-as-a-Judge paradigm (Zheng et al., 2023) uses a large language model (LLM) to evaluate AI system outputs, offering a scalable and efficient alternative to costly and time-consuming human evaluation. Furthermore, beyond merely assessing performance, these evaluators are instrumental in refining models. During training, an LLM-as-a-Judge acts as a scalable reward model to fine-tune performance through automated feedback (Ouyang et al., 2022; Yuan et al., 2024; Luo et al., 2024; Bai et al., 2022), while at inference time, it serves as a real-time filter to evaluate and select the best possible response to eventually form better answers (Faria & Smith, 2025; Lightman et al., 2023).

However, the LLM-as-a-Judge paradigm is undermined by inherent flaws. Judge models are susceptible to biases such as positional (Shi et al., 2024), verbosity (Saito et al., 2023), and self-enhancement (Wataoka et al., 2024), which skew evaluation results and call the paradigm's reliability into question. In response, various benchmarks have been developed to scrutinize the judges themselves (Zheng et al., 2023; Gera et al., 2025; Pu et al., 2025; Chiang et al., 2023). Yet, the methodology of these benchmarks presents its own challenges, as they almost universally rely on human-annotated ground truth. Scoring LLM judges based on their consistency with this human data, particularly on subjective questions, leads to two fundamental issues:

- First, the acquisition of human annotations is a notoriously expensive and labor-intensive process, limiting the scale and diversity of datasets (Horych et al., 2024; Liao et al., 2025).
- Second, and more fundamentally, assuming human judgment as a gold standard is precarious, a *"bitter lesson"* where human-induced biases compromise AI evaluation (Sutton, 2019). As illustrated in Figure 1, this reliance is problematic. Persistent inter-annotator disagreement creates

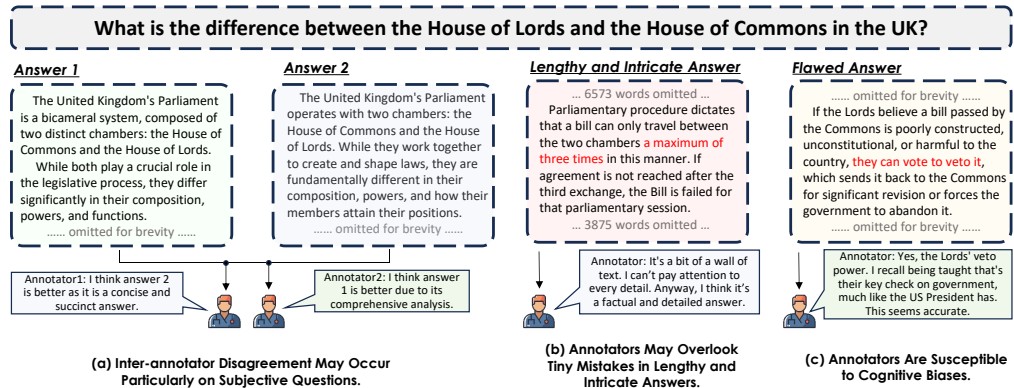

Figure 1: Human-annotated preference may not be reliable. We find three key challenges with relying on human annotators for evaluating LLM-as-a-Judge systems. (a) Inter-annotator Disagreement: Different annotators can have conflicting preferences, especially for subjective questions, leading to noisy and inconsistent data. (b) Overlooking Nuances: Annotators may miss subtle errors or inaccuracies in lengthy and complex answers, leading to flawed evaluations. (c) Cognitive Biases: Human evaluators are susceptible to cognitive biases, such as favoring an answer that confirms their false beliefs, which can further compromise the objectivity of the assessment.

noisy data (Zhang et al., 2024), demonstrated by low agreement shown in AlpacaFarm (66%, Dubois et al. (2023)) and MT-Bench (63%, Zheng et al. (2023)). This problem is compounded when lengthy answers tax human cognitive capacity. Furthermore, human evaluators are susceptible to cognitive biases (Wu & Aji, 2025; Zheng et al., 2023; Chen et al., 2024a), favoring answers that match with their false beliefs, making human annotations an unreliable foundation.

To address this challenge, we introduce SAGE (**S**elf-**A**ssessing **G**auge for **E**valuators), a novel evaluation suite for assessing LLM-as-a-Judge robustness without any human annotation. Our approach is grounded in fundamental principles of rational decision-making, which posit that a reliable judge must exhibit consistent and coherent preferences. For example, a robust judge's preference between two answers should not flip simply because their presentation order is swapped. Furthermore, its judgments should adhere to the principle of transitivity, maintaining a logical and consistent order across a full set of preferences (Ouyang et al., 2022; Song et al., 2024; Hou et al., 2024; Hu et al., 2024; Liu et al., 2024). A breakdown in this coherence suggests the model lacks a consistent internal gauging principle for the question, leading to unreliable situational preferences.

Based on these principles, we propose two metrics to quantify this robustness: **Intra-Pair Instability (IPI)** and **Weak Total Order Violation (TOV)**. IPI directly measures the local, pairwise consistency by detecting instabilities caused by positional bias, as in the first example. TOV, on the other hand, assesses the global logical coherence of a judge's complete set of preferences, identifying systemic contradictions like the violation of transitivity described.

For the evaluation, we curate a diverse dataset of 650 questions by combining selections from RewardBench2 (Gureja et al., 2025) and the large-scale WildChat-1m corpus (Zhao et al., 2024) to ensure broad coverage of real-world user queries. On this dataset, we conduct a comprehensive evaluation of thirteen prominent LLMs. We validate the soundness of our metrics in empirical and theoretical way by consistent checking and a distribution-free error bounding method that quantify the statistical certainty of our results, confirming that the metrics have minuscule variance on the order of $10^{-4}$. A high correlation with established LLMBar (Zeng et al., 2023) and RewardBench2 LLM-as-a-judge benchmark also demonstrates this.

Based on SAGE, we evaluate a wide range of systems, including state-of-the-art LLMs, fine-tuned judges, and multi-agent as juries. All judge models degrade when encountering answers with a close quality gap, with an average decline of 180.1% and 191.0% in IPI and TOV scores, highlighting the potential problem in using LLM-as-a-Judge in RL-based training and test-time scaling. Our findings reveal that current models exhibit significant robustness deficiencies and specialized fine-tuning does not guarantee improvement, as some models' robustness paradoxically degrades by up to 24%. Our findings also show that multi-agent panels can improve performance by up to 12% and that increasing a model's reasoning depth improves logical coherence

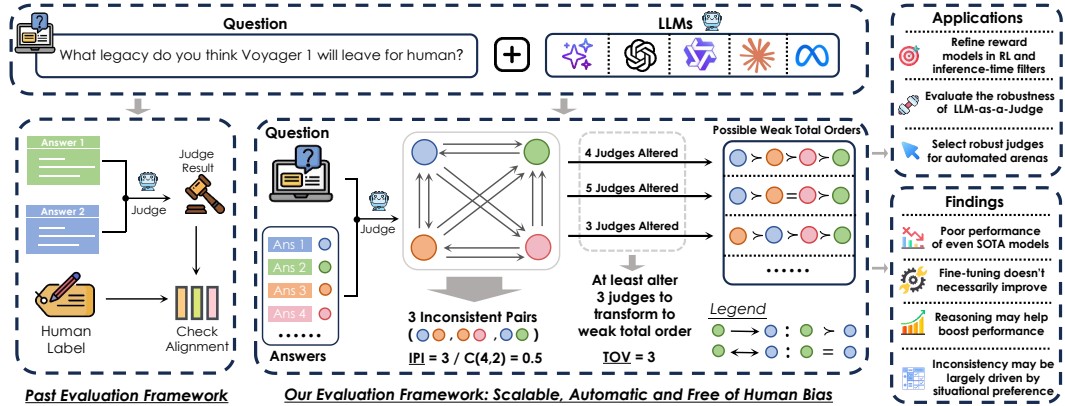

Figure 2: SAGE uses a symmetrized, round-robin protocol to conduct pairwise comparisons on a set of candidate answers. From these judgments, SAGE quantifies performance using two metrics: **IPI**, which measures local consistency by tracking preference reversals (*e.g.*, 3 inconsistent pairs result in an IPI of 0.5), and **TOV**, which assesses global logical coherence by calculating the minimum alterations required for a consistent ranking (*e.g.*, 3 alternations required). This methodology scalably diagnoses logical deficiencies to help identify and select more reliable LLM evaluators.

by over 11%. Notably, prompting for self-generated rubrics to avoid situational preference yields an even greater performance boost, reducing local inconsistency (IPI) and global inconsistency (TOV) by 23.4% and 19.3%, respectively. Lastly, we demonstrate SAGE's practical utility in selecting stable evaluators for automated arenas. We will release all source code, curated dataset at `https://anonymous.4open.science/r/SAGE-6601/`.

## 2 ASSESSING LLM-AS-A-JUDGE WITH SAGE

This section details the foundational methodology of our proposed framework, SAGE. We begin by formally defining the evaluation problem and introducing a symmetrized protocol. Building on this, we then present our two novel metrics: Intra-Pair Instability (IPI) to assess local, pairwise consistency, and Weak Total Order Violation (TOV) to measure global, logical coherence.

### 2.1 PROBLEM FORMULATION

Let $M$ be the LLM under evaluation, referred to as the **judge model**. Our evaluation is based on a set of questions $\mathcal{Q}$. For any given question $Q \in \mathcal{Q}$, we generate a set of $n$ candidate answers, denoted as $A_Q = \{A_1, A_2, \ldots, A_n\}$. The core task of the judge model $M$ is to perform a **pairwise comparison** between any two answers, $A_i$ and $A_j$, from the set $A_Q$. We define a function $J_M$:

$$y_{ij} = J_M(Q, A_i, A_j) \in \{-1, 0, 1\} \tag{1}$$

where the outcome $y_{ij}$ is interpreted as:

- $y_{ij} = 1$: $M$ judges $A_i$ to be superior to $A_j$ ($A_i \succ A_j$).
- $y_{ij} = -1$: $M$ judges $A_i$ to be inferior to $A_j$ ($A_i \prec A_j$).
- $y_{ij} = 0$: $M$ judges $A_i$ and $A_j$ to be of equal quality ($A_i = A_j$).

For each question $Q$, we conduct a full round-robin evaluation, assessing all $\binom{n}{2}$ unique pairs of answers, to establish a complete set of pairwise judgments for our subsequent coherence analysis.

### 2.2 SYMMETRIZED EVALUATION PROTOCOL

A naive single-pass evaluation is susceptible to **positional bias**, where the order of presentation influences the outcome. To substantiate that positional bias does exist in SAGE, we sample 1120 answer pairs and measure the **inconsistent rate** for Llama3-8B-Instruct (Dubey & et al., 2024), Gemini-2.5-Flash-Lite (Comanici & et al., 2025), and Qwen3-4B-Instruct-2507 (Team, 2025).

We define this rate as the frequency of judgments that are not the logical inverse when the answer order is reversed (i.e., $J_M(Q, A_i, A_j) \neq -J_M(Q, A_j, A_i)$). The results in Table 1 confirm the presence of bias. To tackle this issue, we adopt a **symmetrized evaluation protocol**. For each unordered pair of answers $\{A_i, A_j\}$, we query the judge model twice:

Table 1: Local inconsistency (*i.e.*, Positional Bias) across LLM-as-a-Judge.

| Model | Inconsistency (%) |
|---|---|
| Llama3-8B-Instruct | 76.2 |
| Gemini-2.5-Flash-Lite | 25.3 |
| Qwen-3-4B-Instruct | 44.4 |

Forward pass: $y_{ij} \leftarrow J_M(Q, A_i, A_j)$; Reversed pass: $y_{ji} \leftarrow J_M(Q, A_j, A_i)$.

This protocol provides a direct way to measure and account for first-order positional bias.

## 2.3 TWO EVALUATION METRICS

We propose two metrics to quantify the robustness of an LLM judge, targeting two distinct failure modes: local inconsistency on a single pair and global logical incoherence across a set of answers.

**Intra-Pair Instability (IPI).** This metric assesses robustness from an **atomic**, **local** level. It quantifies inconsistencies arising from both systematic positional bias and the inherent stochasticity of the judge model. Under the symmetrized protocol, a perfectly consistent judge would always produce opposite scores for reversed pairs (i.e., $y_{ij} = -y_{ji}$). The IPI score for a given question $Q$ quantifies the deviation from this ideal by calculating the average disagreement across all unique pairs:

$$\text{IPI}(Q) = \frac{1}{\binom{n}{2}} \sum_{1 \leq i < j \leq n} \mathbb{I}(y_{ij} \neq y_{ji}) \tag{2}$$

A higher IPI score indicates a greater degree of local inconsistency of the judge model.

**Weak Total Order Violation (TOV).** This metric assesses robustness from a **global**, **systematic** level. Specifically, it measures the logical coherence of the judge's full set of preferences for a question. A rational judge's preferences should be transitive and form a **weak total order** (i.e., a total order that allows ties). Let $\mathbf{J}_Q = \{y_{ij}\}_{1 \leq i,j \leq n, i \neq j}$ be the set of derived preference from the symmetrized evaluation for a question $Q$. Let $\mathcal{O}_n$ be the set of all possible valid weak total orders on $n$ items. For any order $O \in \mathcal{O}_n$, we can represent it as a corresponding set of pairwise relations $\mathbf{P}_O = \{p_{ij}\}$, where $p_{ij} \in \{-1, 0, 1\}$ denotes the pairwise relationship between items $i$ and $j$ with the order $O$. Specifically, $p_{ij} = 1$ if $i$ is preferred to $j$, $p_{ij} = -1$ if $j$ is preferred to $i$, and $p_{ij} = 0$ if they are tied. The TOV score is defined as the minimum number of preference changes required to transform the judge's observed preferences $\mathbf{P}_Q$ into any valid weak total order:

$$\text{TOV}(Q) = \min_{O \in \mathcal{O}_n} \sum_{1 \leq i,j \leq n, i \neq j} \mathbb{I}(y_{ij} \neq p_{ij}) \tag{3}$$

A higher TOV score signifies more severe logical contradictions in the judge's reasoning.

To summarize a judge model's overall performance, we compute aggregate scores for both IPI and TOV. The aggregate IPI and TOV scores are the arithmetic mean of the per-question scores over the entire set of questions $\mathcal{Q}$ in SAGE, calculated as $\text{IPI} = (1/|\mathcal{Q}|) \sum_{Q \in \mathcal{Q}} \text{IPI}(Q)$ and $\text{TOV} = (1/|\mathcal{Q}|) \sum_{Q \in \mathcal{Q}} \text{TOV}(Q)$. The stability of these metrics is validated empirically in Section 4 and supported by the theoretical analysis in Appendix B.

## 3 THE CONSTRUCTION OF SAGE

We source the question set $\mathcal{Q}$ from five RewardBench2 (Gureja et al., 2025) categories and the large-scale WildChat1M corpus (Zhao et al., 2024) to better reflect real-world user interactions. The resulting question set consists of 650 questions, and its category composition is shown in Figure 3a. To validate its semantic diversity, we use a t-SNE visualization (van der Maaten & Hinton, 2008) to project our questions against a background of 500k English questions from WildChat1M. As shown in Figure 3b, our questions spread broadly across the embedding space, confirming the dataset's representativeness and wide topical coverage. Further details are provided in Appendix C.1.

For each of the 650 questions, we generate a set of $n = 6$ candidate answers for the LLM judge to evaluate, which were used to construct two distinct tiers: SAGE-EASY and SAGE-HARD.

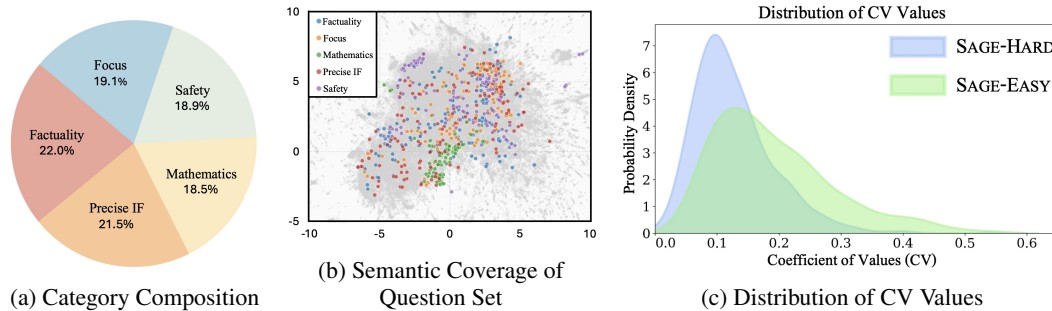

(a) Category Composition

(b) Semantic Coverage of
Question Set

(c) Distribution of CV Values

Figure 3: We provide statistics and analysis of our selected queries and answers within SAGE. Distribution of CV values shows the varied difficulty among our two subsets.

- **SAGE-EASY**: the six answers are generated by a diverse lineup of six models with a clear capability gradient: Gemini-2.5-Pro and Gemini-2.5-Flash (Comanici & et al., 2025); Qwen3-32B (Team, 2025), Claude-3-Haiku (Anthropic, 2024), Llama-3.2-3B-Instruct, and Llama-3.2-1B-Instruct (Meta, 2024b). These models, which have a well-documented performance gap on the LMSYS Chatbot Arena leaderboard (LMSYS, 2025), produce a set of answers with a wide variance in quality, making the pairwise comparison task relatively simple for a competent judge. Moreover, SAGE-EASY reflects the general-purpose task of comparing different models of varying capabilities, which is largely used in automated judges like MT-Bench and Arena Hard Auto.
- **SAGE-HARD**: all six answers for each question are generated by a single capable model, Gemini-2.5-Flash. Since the answers originate from the same model, their quality is expected to be much more homogeneous. This setup presents a more challenging task, requiring the judge to make finer-grained distinctions between subtly different responses. SAGE-HARD models the judge's role in applications like model-based reinforcement learning and rejection sampling. In these scenarios, the judge must distinguish between subtly varied outputs from a single capable model.

To quantitatively confirm the difference in quality diversity between these two tiers, a state-of-the-art reward model, QRM-Gemma-2-27B (Dorka, 2024), is employed to score each of the six answers for every question. For each question, the Coefficient of Variation (CV) of the six reward scores is then calculated. The CV, defined as the ratio of the standard deviation to the mean $(\sigma/\mu)$, is a normalized measure of dispersion. As shown in Figure 3c, the CV distribution for SAGE-HARD is markedly shifted towards lower values, empirically confirming that the answers within its sets are more similar in quality and thus present a more formidable challenge for LLM judges.

## 4 EXPERIMENT AND ANALYSIS

We first conduct a series of validation experiments to prove the internal consistency and external validity of our metrics in Section 4.1 and 4.2. We then employ SAGE to evaluate a diverse set of thirteen *popular* LLMs-as-a-Judge, six specialized fine-tuned judges, and multi-agent configurations. The results highlight significant robustness challenges in *state-of-the-art* LLMs, especially on difficult, fine-grained distinction tasks (Figure 7). Our in-depth analysis reveals that fine-tuning offers no guarantee of improved robustness and that multi-agent judges may boost performance. We attribute it to a new phenomenon we discover, **situational preference**, which can be mitigated by deep reasoning and self-generated rubrics for a more consistent modeling of the question.

### 4.1 VALIDATING METRIC STABILITY AND ROBUSTNESS

A critical aspect of a reliable framework is the stability of its evaluation metrics against the inherent stochasticity of Models. To validate that our proposed metrics are not unduly influenced by random sampling variations, we analyze their stability from both an empirical and a theoretical standpoint. Furthermore, we demonstrate that temperature settings wouldn't threaten the robustness of SAGE.

**Empirical Analysis.** We select two representative models, Qwen3-4B-Instruct-2507 and Qwen3-30B-A3B-Instruct-2507, and evaluate each 50 times on the complete SAGE-EASY and SAGE-HARD. We then calculate the variance of the IPI and TOV scores across these 50 independent

Table 3: Pearson Correlation Coefficients between SAGE metrics and external benchmarks. "Easy-IPI" refers to the IPI metric on SAGE-EASY, and similarly for metric Hard-IPI, Easy-TOV, and Hard-TOV. There is a strong correlation between SAGE and external benchmarks.

| | LLMBar | | | | RewardBench2 | |
|---|---|---|---|---|---|---|
| Easy-IPI | Easy-TOV | Hard-IPI | Hard-TOV | Easy-TOV | Hard-TOV |
| 0.7997 | 0.7904 | 0.7513 | 0.7504 | 0.8705 | 0.7509 |
| 0.7811 | 0.7723 | 0.7301 | 0.7299 | 0.8651 | 0.7488 |

runs. As presented in Table 2, the observed variances are exceptionally low, which provides strong empirical evidence that our metrics are highly reproducible and capture the fundamental reasoning patterns of the judge model rather than ephemeral artifacts of its generative process.

**Theoretical Guarantees.** Our argument proceeds in three stages. First, using principles from Conformal Prediction (Angelopoulos & Bates, 2021), we establish a probabilistic guarantee that any single pairwise judgment, $y_{\text{new}}$, is highly stable and matches its most probable outcome, $y_{\text{new}}^*$, with high confidence:

$$P(y_{\text{new}} = y_{\text{new}}^*) \geq 1 - \alpha. \tag{4}$$

Second, we leverage this result to derive a tight upper bound on the variance of the per-question metrics. For IPI, the score is a fraction of inconsistent pairs out of $N = \binom{6}{2} = 15$ unique pairs. The deviation from the stable score, $\Delta_{\text{IPI}}(Q)$, is bounded by the number of unstable judgments $X$. This allows us to bound the variance as:

$$\text{Var}(\text{IPI}(Q)) \leq \mathbb{E}[\Delta_{\text{IPI}}(Q)^2] \leq \frac{1}{N^2}\mathbb{E}[X^2] \leq \frac{0.387}{15^2} \approx 0.00172. \tag{5}$$

Finally, we show that this variance diminishes over the aggregate evaluation suite. Assuming the per-question scores are independent and identically distributed over our diverse set of $|\mathcal{Q}| = 650$ questions, the variance of the final aggregate IPI score is given by:

$$\text{Var}(\text{IPI}) = \frac{\text{Var}(\text{IPI}(Q))}{|\mathcal{Q}|} \leq \frac{0.00172}{650} \approx 2.65 \times 10^{-6}. \tag{6}$$

A similar derivation establishes an upper bound for Var(TOV) which is $\text{Var}(\text{TOV}) \leq 5.95 \times 10^{-4}$. These theoretical results align perfectly with our empirical findings, confirming that the final reported scores are highly stable. The full derivation of this analysis is available in Appendix B.

**Consistency across Temperatures.** To further validate the stability of SAGE, we evaluate model performance across various temperature settings. The resulting IPI and TOV scores demonstrate remarkable consistency, indicating that our metrics effectively capture the fundamental reasoning capabilities of the models rather than superficial sampling artifacts. For all models and metrics tested, the variance in

Table 2: Variance across 50 independent runs for LLM-as-a-Judge consistency checking.

| Model | Set | IPI Variance | TOV Variance |
|---|---|---|---|
| Qwen3-4B | Easy | $2.9 \times 10^{-7}$ | $6.3 \times 10^{-5}$ |
| | Hard | $8.2 \times 10^{-7}$ | $1.7 \times 10^{-4}$ |
| Qwen3-30B-A3B | Easy | $6.7 \times 10^{-7}$ | $1.5 \times 10^{-4}$ |
| | Hard | $2.1 \times 10^{-6}$ | $4.4 \times 10^{-4}$ |

the scores is less than $4.5 \times 10^{-4}$, which further substantiates the reliability of our framework. More results are presented in Appendix E.1.

## 4.2 VALIDATING SAGE AS A PROXY FOR ROBUSTNESS AND ACCURACY

**Correlation with LLMBar.** To establish the credibility of SAGE as a new evaluation framework, we first validate its external alignment with existing methodologies by comparing our robustness metrics against LLMBar (Zeng et al., 2023), an established benchmark that evaluates LLM-as-a-Judge systems using human-annotated ground truth. We focus specifically on the adversarial subset of LLMBar, which is designed to stress-test the robustness of judge models. This subset contains instances where one response is correct while the other is adversarially crafted to be superficially appealing, thus challenging a judge's ability to remain robust against deceptive quality. We test the same thirteen models evaluated in Section 4.3 on both SAGE and the LLMBar adversarial subset. As shown in Table 3, the results reveal a strong positive correlation between the models' error rates

Table 4: The performance of thirteen LLMs on SAGE, with lower scores indicate greater robustness. A clear trend emerges where advanced models like Gemini-2.5-Pro demonstrate superior robustness.

| Models | Factuality | | Precise IF | | Mathematics | | Safety | | Focus | | Overall | |
|---|---|---|---|---|---|---|---|---|---|---|---|---|
| | IPI↓ | TOV↓ | IPI↓ | TOV↓ | IPI↓ | TOV↓ | IPI↓ | TOV↓ | IPI↓ | TOV↓ | IPI↓ | TOV↓ |
| *Performance on* SAGE-EASY | | | | | | | | | | | | |
| Gemini-2.5-Pro | **0.064** | **0.993** | **0.091** | **1.367** | **0.071** | **1.135** | 0.123 | 1.942 | **0.062** | **0.927** | **0.082** | **1.265** |
| Gemini-2.5-Flash | 0.077 | 1.175 | 0.133 | 2.043 | 0.082 | 1.305 | 0.105 | 1.667 | 0.075 | 1.137 | 0.095 | 1.471 |
| Qwen3-235B-A22B-Instruct-2507 | 0.077 | 1.175 | 0.117 | 1.761 | 0.150 | 2.310 | **0.101** | **1.626** | 0.091 | 1.374 | 0.106 | 1.626 |
| Qwen3-4B-Instruct-2507 | 0.110 | 1.664 | 0.151 | 2.288 | 0.166 | 2.492 | 0.130 | 1.992 | 0.090 | 1.347 | 0.129 | 1.952 |
| DeepSeek-V3-0324 | 0.105 | 1.601 | 0.141 | 2.108 | 0.194 | 3.058 | 0.115 | 1.821 | 0.094 | 1.417 | 0.129 | 1.989 |
| DeepSeek-V3.1 | 0.107 | 1.645 | 0.160 | 2.425 | 0.172 | 2.780 | 0.159 | 2.451 | 0.109 | 1.683 | 0.141 | 2.182 |
| DeepSeek-R1-0528 | 0.114 | 1.725 | 0.189 | 2.914 | 0.147 | 2.421 | 0.154 | 2.424 | 0.104 | 1.593 | 0.142 | 2.222 |
| GPT-5-Chat | 0.111 | 1.671 | 0.226 | 3.389 | 0.132 | 2.108 | 0.132 | 2.008 | 0.157 | 2.379 | 0.152 | 2.319 |
| GPT-4o-mini | 0.114 | 1.706 | 0.144 | 2.179 | 0.239 | 3.600 | 0.184 | 2.959 | 0.088 | 1.331 | 0.152 | 2.323 |
| Qwen3-30B-A3B-Instruct-2507 | 0.135 | 2.035 | 0.125 | 1.893 | 0.190 | 2.850 | 0.332 | 5.008 | 0.135 | 2.024 | 0.180 | 2.715 |
| Gemini-2.0-Flash-Lite | 0.152 | 2.280 | 0.179 | 2.686 | 0.224 | 3.375 | 0.247 | 3.878 | 0.164 | 2.460 | 0.191 | 2.906 |
| Claude-3-Haiku | 0.225 | 3.392 | 0.342 | 5.138 | 0.323 | 4.908 | 0.396 | 5.984 | 0.201 | 3.048 | 0.296 | 4.468 |
| Llama-3.1-8B-Instruct | 0.360 | 5.640 | 0.353 | 5.625 | 0.406 | 6.475 | 0.341 | 5.261 | 0.358 | 5.554 | 0.364 | 5.710 |
| *Performance on* SAGE-HARD | | | | | | | | | | | | |
| Gemini-2.5-Pro | 0.277 | 4.490 | **0.290** | **4.600** | **0.133** | **2.517** | 0.249 | 4.276 | 0.317 | 5.169 | **0.244** | **4.239** |
| Gemini-2.5-Flash | **0.269** | **4.091** | 0.316 | 4.864 | 0.223 | 3.983 | **0.233** | **3.984** | **0.278** | **4.420** | 0.266 | 4.280 |
| DeepSeek-V3-0324 | 0.381 | 5.921 | 0.351 | 5.393 | 0.277 | 4.740 | 0.309 | 4.901 | 0.418 | 6.484 | 0.349 | 5.504 |
| Qwen3-235B-A22B-Instruct-2507 | 0.382 | 6.126 | 0.325 | 4.986 | 0.285 | 4.824 | 0.297 | 5.211 | 0.457 | 7.282 | 0.350 | 5.691 |
| Qwen3-4B-Instruct-2507 | 0.388 | 5.846 | 0.372 | 5.586 | 0.324 | 5.083 | 0.390 | 5.886 | 0.455 | 6.855 | 0.386 | 5.849 |
| GPT-4o-mini | 0.436 | 6.993 | 0.458 | 7.086 | 0.337 | 5.375 | 0.358 | 5.724 | 0.487 | 7.992 | 0.417 | 6.665 |
| DeepSeek-V3.1 | 0.486 | 7.979 | 0.522 | 8.093 | 0.174 | 3.250 | 0.382 | 6.309 | 0.489 | 8.460 | 0.417 | 6.905 |
| GPT-5-Chat | 0.467 | 7.196 | 0.581 | 8.800 | 0.191 | 3.250 | 0.352 | 5.650 | 0.615 | 9.331 | 0.447 | 6.928 |
| DeepSeek-R1-0528 | 0.432 | 7.200 | 0.493 | 8.157 | 0.203 | 3.757 | 0.408 | 6.813 | 0.501 | 8.618 | 0.413 | 6.993 |
| Gemini-2.0-Flash-Lite | 0.656 | 9.902 | 0.565 | 8.521 | 0.443 | 6.842 | 0.318 | 5.236 | 0.745 | 11.371 | 0.550 | 8.437 |
| Claude-3-Haiku | 0.552 | 8.469 | 0.578 | 8.797 | 0.551 | 9.183 | 0.539 | 8.545 | 0.574 | 8.734 | 0.559 | 8.736 |
| Llama-3.1-8B-Instruct | 0.555 | 8.706 | 0.518 | 7.907 | 0.706 | 10.725 | 0.789 | 11.968 | 0.586 | 9.040 | 0.625 | 9.588 |
| Qwen3-30B-A3B-Instruct-2507 | 0.647 | 9.699 | 0.440 | 6.614 | 0.637 | 9.775 | 0.785 | 11.772 | 0.765 | 11.476 | 0.649 | 9.780 |

on LLMBar and our proposed metrics. This strong statistical alignment validates that SAGE serves as a reliable proxy for judging model robustness without the need for costly manual annotation.

**Proxy for Accuracy.** Beyond robustness, we argue that SAGE can also function as an effective proxy for judging accuracy. Theoretically, TOV quantifies the minimum number of pairwise judgments that must be altered for the entire set to become logically coherent. Since logical coherence is a prerequisite for correctness, the total number of errors in a set of judgments must be at least as large as the minimum alterations needed to resolve its logical contradictions. Therefore, TOV establishes a rigorous **lower bound** on the error rate. To empirically substantiate this claim, we leverage a 599-question subset of our evaluation suite for which ground-truth preference labels are available from the RewardBench2. We evaluate the same thirteen LLMs, calculating each model's error rate against the provided ground-truth and comparing it with their TOV scores from SAGE. As shown in Table 3, we see a significantly high Pearson Correlation between the models' ground-truth error rates and their TOV scores, proving that SAGE can serve as a robust proxy for judgment accuracy.

## 4.3 EVALUATING LLM-AS-A-JUDGE WITH SAGE

We benchmark thirteen popular LLMs with the aforementioned settings, including five proprietary models (*i.e.* Gemini-2.5-Pro and Gemini-2.5-Flash (Comanici & et al., 2025); Gemini-2.0-Flash-Lite (Google, 2025), GPT-5-Chat (OpenAI, 2025), GPT-4o-mini (OpenAI, 2024) and Claude-3-Haiku (Anthropic, 2024)) and seven open source models (*i.e.* Qwen3-235B-A22B-Instruct-2507, Qwen3-30B-A3B-Instruct-2507 and Qwen3-4B-Instruct-2507 (Team, 2025); DeepSeek-R1-0528 (DeepSeek-AI, 2025a), DeepSeek-V3 (DeepSeek-AI & et al., 2024), DeepSeek-V3.1 (DeepSeek-AI, 2025b), Llama-3.1-8B-Instruct (Meta, 2024a)). The results are shown in Table 4. All evaluations are conducted at the default temperature to ensure a fair and consistent comparison.

Table 5: Our experiments on SAGE-HARD show that specialized finetuned judges varies in their stability, with some even falling behind its base model.

| Models | Factuality | | Precise IF | | Mathematics | | Safety | | Focus | | Overall | |
|---|---|---|---|---|---|---|---|---|---|---|---|---|
| | IPI↓ | TOV↓ | IPI↓ | TOV↓ | IPI↓ | TOV↓ | IPI↓ | TOV↓ | IPI↓ | TOV↓ | IPI↓ | TOV↓ |
| *Qwen2.5-3B-Instruct (Base)* | 0.637 | 10.206 | 0.565 | 9.092 | 0.559 | 9.000 | 0.603 | 9.703 | 0.571 | 9.513 | 0.586 | 9.483 |
| M-Prometheus-3B | 0.723 | 10.909 | 0.580 | 8.814 | 0.659 | 10.075 | 0.696 | 10.626 | 0.686 | 10.387 | 0.668(↑14%) | 10.151(↑7%) |
| JudgeLRM-3B | 0.823 | 12.371 | 0.610 | 9.171 | 0.801 | 12.025 | 0.826 | 12.382 | 0.869 | 13.057 | 0.782(↑33%) | 11.751(↑24%) |
| *Qwen2.5-7B-Instruct (Base)* | 0.876 | 13.140 | 0.811 | 12.164 | 0.926 | 13.900 | 0.786 | 12.000 | 0.944 | 14.161 | 0.867 | 13.049 |
| M-Prometheus-7B | 0.580 | 8.762 | 0.509 | 7.821 | 0.677 | 10.283 | 0.569 | 8.878 | 0.613 | 9.347 | 0.587(↓32%) | 8.974(↓24%) |
| JudgeLRM-7B | 0.939 | 14.077 | 0.883 | 13.243 | 0.960 | 14.400 | 0.976 | 14.634 | 0.973 | 14.589 | 0.944(↑9%) | 14.160(↑9%) |
| *Mistral-7B-Instruct (Base)* | 0.734 | 11.078 | 0.582 | 8.978 | 0.806 | 12.133 | 0.655 | 9.854 | 0.786 | 11.839 | 0.710 | 10.736 |
| Prometheus-7B-V2.0 | 0.616 | 9.634 | 0.546 | 8.773 | 0.602 | 10.000 | 0.553 | 9.186 | 0.652 | 10.105 | 0.592(↓17%) | 9.509(↓11%) |
| *Llama-3.1-8B-Instruct (Base)* | 0.555 | 8.706 | 0.518 | 7.907 | 0.706 | 10.725 | 0.789 | 11.968 | 0.586 | 9.040 | 0.625 | 9.588 |
| Skywork-Critic-Llama-3.1-8B | 0.503 | 7.539 | 0.421 | 6.314 | 0.380 | 5.700 | 0.348 | 5.268 | 0.584 | 8.766 | 0.449(↓28%) | 6.740(↓30%) |

Our comprehensive benchmarking reveals significant robustness deficiencies in current state-of-the-art LLMs. A clear trend emerges where more advanced models, such as Gemini-2.5-Pro, consistently demonstrate superior robustness with the lowest IPI and TOV scores, indicating stronger local self-consistency and global logical coherence. Crucially, all models show a marked degradation in performance from SAGE-EASY to SAGE-HARD with a 180.1% and 191% decline on IPI and TOV scores. This performance gap underscores a key limitation: while models may appear relatively reliable when judging answers of clearly different quality, their adjudicative abilities falter when faced with subtle distinctions, posing a serious threat to their effectiveness in inference-time enhancement techniques like rejection sampling or Monte Carlo Tree Search. These findings highlight that fundamental consistency remains a substantial challenge for LLMs acting as judges.

## 4.4 IN-DEPTH ANALYSIS

**Injustice Judges or Situational Preference?** We argue that a robust LLM-as-a-Judge should first model the question internally regardless of how the answers vary. However, the extremely high IPI and TOV scores across even *state-of-the-art* models raise the concern of whether models are incapable of providing just judgments, or whether their judgments are merely **situational preferences** (Laine et al., 2024; Needham et al., 2025), *i.e.*, inconsistent judging criteria encountering different answers under the same question. To validate this hypothesis, we investigate whether an LLM can improve its evaluation by first explicitly articulating its judging rubrics and then using the rubrics to judge the answers across different judging pairs under the same questions. Crucially, this rubric is generated only once per question and serves as a fixed standard for all answer pairs, a method designed to mitigate situational preferences by preventing the judge's evaluation criteria from shifting between comparisons. Figure 4 shows that this approach yields a notable performance boost, reducing IPI and TOV scores by 23.4% and 19.3%. This gap demonstrates that current LLM-as-a-Judge systems indeed exhibit extreme situational preferences when encountering different answer pairs, and that explicit judging rubrics can substantially mitigate this.

**Do fine-tuned Judges make better judgments?** A fine-tuned judge is an LLM trained on a preference dataset to improve their evaluation. We benchmark six fine-tuned judges (*i.e.* Prometheus-7B-V2.0 (Kim et al., 2024), Skywork-Critic-Llama-3.1-8B (Shiwen et al., 2024), M-Prometheus-3/7B (Pombal et al., 2025), and JudgeLRM-3/7B (Chen et al., 2025)) and their corresponding base models. The results are shown in Table 5. Additional results of their performance on SAGE-EASY are available in Appendix E.2. Our result reveals a mixed impact from fine-tuning. While some models like Skywork-Critic-Llama-3.1-8B show marked improvement, others, particularly the JudgeLRM series, paradoxically become less reliable. We attribute the degradation to biases inherited from the training datasets, which can cause the model to learn and amplify flawed judgment patterns, compromising its objectivity. See Appendix F.4 for the examples of human biases in training data.

**Do Multi-agent Debates or Panels Judge Better?** In our evaluation, we also explore the effectiveness of multi-agent judge systems, an approach intended to reduce bias and improve evaluation robustness. We investigate two distinct methodologies: a panel-based approach inspired by POLL (Verga et al., 2024), which leverages a diverse jury of different LLMs, and a debate-based frame-

Table 6: Performance comparison of multi-agent systems: POLL panels (left) and ChatEval debates (right). For POLL, "Best Indi." refers to the best individual model in the panel.

| Method | IPI-Easy ↓ | TOV-Easy ↓ | IPI-Hard ↓ | TOV-Hard ↓ |
|---|---|---|---|---|
| **Panel 1 (Powerful Models)** | | | | |
| Best Indi. | 0.082 | 1.265 | **0.244** | 4.239 |
| Aggregate | **0.073** (↓11%) | **1.146** (↓9%) | 0.248 (↑2%) | **4.088** (↓4%) |
| **Panel 2 (Weaker Models)** | | | | |
| Best Indi. | 0.141 | 2.182 | 0.417 | 6.665 |
| Aggregate | **0.122** (↓13%) | **1.909** (↓13%) | **0.367** (↓12%) | **5.965** (↓11%) |

| Method | IPI-Easy ↓ | TOV-Easy ↓ | IPI-Hard ↓ | TOV-Hard ↓ |
|---|---|---|---|---|
| **Qwen3-4B-Instruct-2507** | | | | |
| Baseline | **0.129** | **1.952** | **0.386** | **5.849** |
| ChatEval | 0.334 (↑158%) | 5.105 (↑162%) | 0.651 (↑69%) | 10.050 (↑72%) |
| **Qwen3-30B-A3B-Instruct-2507** | | | | |
| Baseline | **0.180** | **2.715** | 0.649 | 9.780 |
| ChatEval | 0.261 (↑45%) | 4.080 (↑50%) | **0.518** (↓20%) | **8.395** (↓14%) |

work, ChatEval (Chan et al., 2023), which utilizes multiple agents derived from a single LLM. The results are shown in Table 6. For the panel approach, we construct two separate juries: the first comprised of powerful models (Gemini-2.5-Pro, Gemini-2.5-Flash, and Qwen3-235B-A22B-Instruct-2507), while the second uses weaker models (Gemini-2.0-Flash-Lite, GPT-4o-mini, and DeepSeek-V3.1). For the POLL method, the aggregated judgments in the majority of cases surpass the performance of the best individual model within each respective group, demonstrating a clear performance boost. Conversely, debate-based ChatEval framework fails to yield an improvement in evaluation quality, demonstrating less robust performance.

**Does Deep Reasoning Lead to Better Performance?** We analyze the distinct effects of a model's intrinsic reasoning depth. For this experiment, we employ the gpt-oss model family (20B and 120B) (Agarwal et al., 2025), for its configurable reasoning modes: low, medium and high. As illustrated in Figure 4, the results show an improvement as the reasoning mode is intensified from low to high.

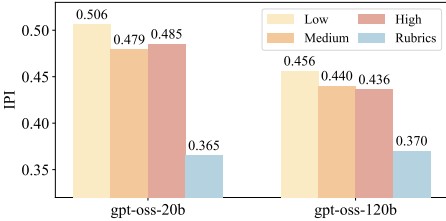 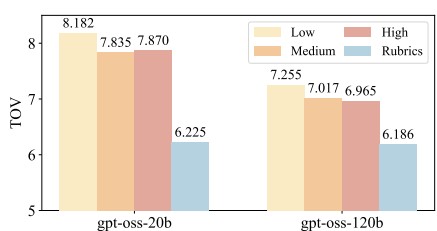

Figure 4: We discover high IPI and TOV scores in SAGE-HARD due to situational preference phenomenon in LLM-as-a-Judge, while deep thinking and explicit rubrics can mitigate this.

**Selecting Robust Judges for Automated Arenas.** Here we explore the practical utility of our framework in selecting robust evaluators for large-scale, automated model ranking systems like Arena-Hard-Auto (Chiang et al., 2024). In such systems, models are ranked using Elo ratings derived from pairwise comparisons. The confidence interval of a model's Elo rating serves as a crucial indicator of its judgment stability; a smaller interval suggests more consistent and more reliable evaluation performance. Our investigation reveals a strong positive correlation between our metrics and the Elo rating confidence intervals from Arena-Hard-Auto. Our IPI and TOV scores show strong Pearson correlations of **0.7638** and **0.7600**, respectively, with the confidence intervals. This strong alignment demonstrates that SAGE can effectively identify more stable judges, making it a valuable tool for selecting high-quality evaluators to enhance the reliability of automated arena rankings.

## 5 CONCLUSION

We introduce SAGE, a novel framework to evaluate LLM-as-a-Judge without human annotation or any extrinsic information by measuring local and global logical consistency. Our experiments reveal significant robustness deficiencies in current state-of-the-art models, demonstrating that fine-tuning can amplify inherited biases and that model diversity is critical for multi-agent evaluators. We validate that our metrics are exceptionally stable and can serve as a strong proxy for accuracy. Consequently, SAGE provides a scalable, reliable, and cost-effective tool to diagnose and improve LLM evaluators, paving the way for more consistent and rational AI systems.

ETHICS STATEMENT

Our dataset is curated from established public research sources: the RewardBench2 benchmark and the WildChat-1m corpus. To mitigate ethical risks, such as the potential inclusion of private information or inappropriate content from real-world user logs, we conducted a rigorous curation process (see Appendix C.1). This process involved both large-scale automated filtering and a thorough manual review of every selected question. This ensures that the final dataset is appropriate for research use and aligns with the data-sharing and privacy standards of the original sources.

REPRODUCIBILITY STATEMENT

To ensure the reproducibility of our research, we will release all source code, the curated dataset, and the collected model responses. The foundational methodology of our framework, including the formal problem definition, the symmetrized evaluation protocol, and the definitions of our IPI and TOV metrics, is detailed in Section 2. The comprehensive process for curating our 650-question dataset is described in Section 3, with further implementation details provided in Appendix C.1. For our theoretical claims, a complete derivation of the variance bounds for our metrics is available in Appendix B. Furthermore, all detailed experimental setups, including descriptions of the models evaluated (Appendix C.4) and the exact prompts used in our experiments, are provided in appendix F to facilitate the replication of our results.

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

## THE USE OF LARGE LANGUAGE MODELS (LLMS)

The use of large language models (LLMs) in this work is strictly limited to auxiliary text editing, such as correcting spelling and improving grammar, and dataset generation. Our study is about LLM-as-a-Judge, therefore we also test various LLMs for this task. All conceptual and technical contributions are the original work of the authors. We are transparent about this limited usage.

## A  RELATED WORK

**LLM-as-a-Judge.**  LLM-as-a-Judge (Zheng et al., 2023) has emerged as a scalable and cost-effective alternative to human evaluation for assessing the quality of generative AI outputs. This approach utilizes a powerful LLM to judge the responses of other models, addressing the limitations of traditional metrics like BLEU and ROUGE that often fail to capture deeper semantic qualities such as coherence, factual accuracy, and relevance.

However, the reliability of LLM-as-a-Judge is a significant concern, with numerous studies (Zheng et al., 2023; Wu & Aji, 2025; Chen et al., 2024a) highlighting its susceptibility to various biases. These include verbosity bias, where longer answers are favored irrespective of their quality; position bias, a preference for the first or last presented response; and self-enhancement bias, where a model tends to rate its own outputs more favorably. Research (Chen et al., 2024a) has also identified other distorting influences, such as authority bias, where an LLM may favor answers containing citations even if they are fabricated. These identified biases underscore the necessity for continued investigation and validation of the reliability of LLM-as-a-Judge.

**Benchmark for LLM-as-a-Judge.**  Following the recognition of these potential biases of LLM-as-a-Judge, researchers have focused on developing specialized benchmarks to systematically evaluate the reliability and behavior of LLM judges. Unlike general-purpose LLM benchmarks that assess broad capabilities, these targeted frameworks are designed specifically to scrutinize the adjudicative performance of models. For instance, foundational benchmarks such as MT-Bench and Chatbot Arena (Zheng et al., 2023) are introduced to verify the agreement between LLM judges and human preferences on open-ended, multi-turn questions. Subsequent works like Tan et al. (2025) and Gera et al. (2025) continue to follow this paradigm, primarily assessing the capability of LLM judges by measuring the correlation between their assessments and human preference judgments.

However, this reliance on human judgment as the definitive *"gold standard"* is unreliable for three key reasons: First, human annotators are susceptible to inherent biases (Wu & Aji, 2025; Zheng et al., 2023), including authority bias and misinformation oversight bias (Chen et al., 2024a). In addition, Chen et al. (2024a) shows that human evaluators of LLMs can be more biased than the models themselves. Second, there is an persistent issue of inter-annotator disagreement (Zhang et al., 2024). Different human evaluators often provide inconsistent assessments, particularly for tasks that are subjective or nuanced. This lack of consensus means that the *"ground truth"* data used for benchmarking is often noisy and unreliable. Finally, as AI models advance, they are beginning to surpass human capabilities in specialized domains. When AI generates highly complex or lengthy outputs, human annotators might struggle to accurately assess their quality or correctness (Tan et al., 2025). In such scenarios, human annotations may no longer be a reliable ground truth.

**Fine-tuned Judge.**  In the pursuit of improving automated evaluation accuracy, one prevalent strategy involves specializing a model using preference datasets, resulting in a fine-tuned *"judge"* model (Zhu et al., 2025; Kim et al., 2024; Wang et al., 2024b;a; He et al., 2024). These datasets generally comprise a series of prompts, each followed by multiple model-generated responses, with evaluators providing labels to indicate the superior response. By leveraging this data, the judge model is trained to predict human evaluative behaviors, enabling it to autonomously score or rank new model outputs. The fine-tuning process allows the judge to learn nuanced patterns in human preferences, such as understanding which aspects of a response are prioritized. As a result, the judge can offer an automated alternative to human evaluation, making it invaluable for large-scale applications where human assessment may be time-consuming or impractical. However, this approach is not without its limitations (Huang et al., 2024a;b). For those judge models that are fine-tuned on datasets derived from human evaluations, they inevitably inherit the biases and inconsistencies present in the human labeling process. Human annotators, despite their best efforts, may display subjective tendencies,

varying interpretation of instructions, or inconsistencies in rating, which can be subtly reflected in the model's predictions (Chen et al., 2024b). As a consequence, the fine-tuned judge may sometimes generate evaluations that do not align with a broader, more objective standard (Gao et al., 2023). Given these challenges, the reliability and fairness of fine-tuned judge models as objective evaluators must be subjected to thorough scrutiny. It becomes crucial to investigate the degree to which these models mirror human biases and assess their robustness across diverse contexts and response types.

## B    THEORETICAL ANALYSIS OF METRIC STABILITY

In this section, we provide a theoretical analysis to substantiate the empirical stability of our proposed metrics, Intra-Pair Instability (IPI) and Weak Total Order Violation (TOV), as presented in 4.1. The core of our analysis is to demonstrate that the variance of these metrics is exceptionally low, thereby ensuring their reliability against the inherent stochasticity of LLM judges.

The foundational source of any potential instability in our evaluation framework stems from the stochastic nature of the LLM judge, $\mathcal{M}$. When queried multiple times with the identical input triplet $(Q, A_i, A_j)$, the model's judgment, $y_{ij} = J_{\mathcal{M}}(Q, A_i, A_j)$, may fluctuate. Our analysis proceeds in three stages: first, we certify the stability of a single pairwise judgment; second, we bound the variance of the per-question metrics; and third, we establish the stability of the final, aggregate benchmark scores.

### B.1    CERTIFYING SINGLE-PAIR JUDGMENT STABILITY VIA CONFORMAL PREDICTION

To formally quantify the stability of individual judgments, we adopt principles from Conformal Prediction (Angelopoulos & Bates, 2021). We posit that for any given question-answer pair, there exists a *"stable judgment"*, which represents the most stable outcome if the model were to be sampled repeatedly. We approximate this stable judgment by the modal outcome over a large number of trials.

We construct a large-scale calibration set, $\mathcal{C}$, by selecting $N = 30,000$ distinct question-answer pairs. For each pair $k \in \{1, \ldots, N\}$, we prompt the LLM judge $T = 20$ times, yielding a total of $N \times T = 600,000$ individual judgments. For each pair $k$, we define its stable judgment, $y_k^*$, as the most frequently observed outcome:

$$y_k^* = \arg\max_{y \in \{-1,0,1\}} \sum_{t=1}^{T} \mathbb{I}(y_k^{(t)} = y)$$

where $y_k^{(t)}$ is the outcome of the $t$-th judgment for the $k$-th pair.

We can now use the $n = 600,000$ judgments in $\mathcal{C}$ to build a calibration set for a new judgment. Let the non-conformity score for a given judgment $y_k^{(t)}$ be its disagreement with the stable judgment: $s(y_k^{(t)}) = \mathbb{I}(y_k^{(t)} \neq y_k^*)$. By applying the conformal prediction framework to this large calibration set of scores, we can construct a prediction interval for a new, unseen judgment. Our empirical analysis on this calibration set reveals that the fraction of judgments deviating from their stable counterpart is exceedingly small. Following the standard procedure for conformal calibration, we can formally certify that for any new judgment $y_{new}$, the probability of it matching its corresponding stable judgment $y_{new}^*$ is bounded with high confidence. Specifically, for a desired miscoverage rate $\alpha = 0.01$, the procedure yields the following guarantee:

$$P(y_{new} = y_{new}^*) \geq 1 - \alpha = 0.99$$

This result provides a strong probabilistic guarantee that any single pairwise comparison performed by the judge is highly likely to be stable.

### B.2    BOUNDING THE VARIANCE OF PER-QUESTION METRICS

For each question in our benchmark, the calculation of IPI and TOV scores relies on a set of pairwise comparisons. Given that we generate $n = 6$ candidate answers, a full round-robin evaluation under

our symmetrized protocol requires $M = 2 \times \binom{6}{2} = 30$ individual judgments. Our objective is to establish a rigorous, high-confidence upper bound for the variance of the per-question metric, $\text{Var}(\text{TOV}(Q))$, which arises from the LLM judge's inherent stochasticity.

By definition, the variance of the measured score $\text{TOV}(Q)$ is the expected squared difference from its mean:

$$\text{Var}(\text{TOV}(Q)) = \mathbb{E}\left[(\text{TOV}(Q) - \mathbb{E}[\text{TOV}(Q)])^2\right] \tag{7}$$

A fundamental property of variance is that it represents the minimum possible expected squared error. For any constant $c$, the following inequality holds: $\text{Var}(\text{TOV}(Q)) \leq \mathbb{E}\left[(\text{TOV}(Q) - c)^2\right]$. We can leverage this property by strategically choosing a constant. Let us choose the deterministic stable score, $\text{TOV}^*(Q)$, as our constant $c$. This yields this inequality:

$$\text{Var}(\text{TOV}(Q)) \leq \mathbb{E}\left[(\text{TOV}(Q) - \text{TOV}^*(Q))^2\right] \tag{8}$$

Let the deviation from the stable score be $\Delta_{\text{TOV}}(Q) = \text{TOV}(Q) - \text{TOV}^*(Q)$. Equation 8 can be rewritten as:

$$\text{Var}(\text{TOV}(Q)) \leq \mathbb{E}[\Delta_{\text{TOV}}(Q)^2] \tag{9}$$

Our task now simplifies to finding an upper bound for the second moment of this deviation.

Let $X$ be the random variable for the total number of unstable judgments among the $M = 30$ trials. As established in Section B.1, the probability $p$ of any single judgment being unstable is bounded by $p \leq \alpha = 0.01$. Assuming independence across judgments, $X$ follows a Binomial distribution, $X \sim \mathcal{B}(M, p)$.

A direct, deterministic relationship connects the score deviation to the number of unstable judgments. Since the TOV score is the minimum number of edge modifications required to resolve all logical contradictions, $X$ unstable judgments can alter the final score by at most $X$. This gives the inequality $|\Delta_{\text{TOV}}(Q)| \leq X$, which implies:

$$\Delta_{\text{TOV}}(Q)^2 \leq X^2 \tag{10}$$

By taking the expectation, we can chain the inequalities together:

$$\text{Var}(\text{TOV}(Q)) \leq \mathbb{E}[\Delta_{\text{TOV}}(Q)^2] \leq \mathbb{E}[X^2] \tag{11}$$

The second moment of a binomial random variable is given by $\mathbb{E}[X^2] = \text{Var}(X) + (\mathbb{E}[X])^2 = Mp(1-p) + (Mp)^2$. Using $M = 30$ and the upper bound $p = 0.01$, we compute:

$$\mathbb{E}[X] = 30 \times 0.01 = 0.3 \tag{12}$$
$$\text{Var}(X) = 30 \times 0.01 \times (1 - 0.01) = 0.297 \tag{13}$$

Therefore, the second moment of $X$ is:

$$\mathbb{E}[X^2] = 0.297 + (0.3)^2 = 0.387 \tag{14}$$

This directly provides a tight and rigorously derived upper bound for the variance of the per-question TOV score:

$$\text{Var}(\text{TOV}(Q)) \leq 0.387 \tag{15}$$

This result formally demonstrates that the variance of the per-question scores is exceptionally small, confirming that our metrics are highly robust to the inherent stochasticity of LLM judges.

An identical argument holds for the IPI score, yielding a similarly small per-question variance. The IPI score for a question, $\text{IPI}(Q)$, is the fraction of inconsistent pairs. It is calculated over $N = \binom{6}{2} = 15$ unique pairs of answers. Each inconsistent pair contributes 1 to a sum, which is then normalized by $N$. An unstable judgment can affect the consistency of at most one pair, thus changing the sum by at most 1. Therefore, $X$ unstable judgments can change the sum of inconsistent pairs by at most $X$. The deviation of the normalized IPI score, $\Delta_{\text{IPI}}(Q)$, is thus bounded by:

$$|\Delta_{\text{IPI}}(Q)| \leq \frac{X}{N} \tag{16}$$

It is worth noting that this inequality can be tightened; since the IPI score is bounded in $[0, 1]$, the maximal deviation is 1, making the true bound $|\Delta_{\text{IPI}}(Q)| \leq \min(X/N, 1)$. By proceeding with the

analytically simpler $X/N$, we are establishing a conservative overestimate for the variance, which strengthens our claim of stability. Following the same logic, we can bound its variance:

$$\text{Var}(\text{IPI}(Q)) \leq \mathbb{E}[\Delta_{\text{IPI}}(Q)^2] \leq \mathbb{E}\left[\left(\frac{X}{N}\right)^2\right] = \frac{1}{N^2}\mathbb{E}[X^2] \tag{17}$$

Substituting $N = 15$ and our previously calculated value for $E[X^2]$:

$$\text{Var}(\text{IPI}(Q)) \leq \frac{0.387}{15^2} = \frac{0.387}{225} \approx 0.00172 \tag{18}$$

These results formally demonstrate that the variances of both per-question TOV and IPI scores are exceptionally small, confirming that our metrics are highly robust to the inherent stochasticity of LLM judges.

### B.3 STABILITY OF AGGREGATE BENCHMARK SCORES

The final SAGE metrics are the aggregate scores, IPI and TOV, which are the arithmetic means of the per-question scores over the entire set of $|\mathcal{Q}| = 650$ questions:

$$\text{TOV} = \frac{1}{|\mathcal{Q}|}\sum_{Q \in \mathcal{Q}}\text{TOV}(Q) \quad \text{and} \quad \text{IPI} = \frac{1}{|\mathcal{Q}|}\sum_{Q \in \mathcal{Q}}\text{IPI}(Q) \tag{19}$$

Assuming the scores for each question are independent and identically distributed (i.i.d.) random variables—a standard assumption for a diverse benchmark—the variance of the mean is the per-question variance divided by the number of questions.

Using the upper bound for the per-question TOV variance derived in Section B.2, we can bound the variance of the final aggregate TOV score:

$$\text{Var}(\text{TOV}) = \frac{\text{Var}(\text{TOV}(Q))}{|\mathcal{Q}|} \leq \frac{0.387}{650} \approx 5.95 \times 10^{-4} \tag{20}$$

Similarly, using the upper bound for the per-question IPI variance, we can bound the variance of the final aggregate IPI score:

$$\text{Var}(\text{IPI}) = \frac{\text{Var}(\text{IPI}(Q))}{|\mathcal{Q}|} \leq \frac{0.00172}{650} \approx 2.65 \times 10^{-6} \tag{21}$$

These resulting variances for both aggregate metrics are exceptionally small, indicating that the final reported scores are highly concentrated around their expected values.

In conclusion, this theoretical analysis, grounded in first principles and basic statistical properties, formally demonstrates the robustness of our evaluation framework. The high stability of individual judgments propagates through the metric calculation, resulting in aggregate scores for both IPI and TOV with minimal variance. This theoretical finding is in strong alignment with the empirical results presented in Table 2, confirming that SAGE provides a consistent and reliable methodology for assessing the reasoning capabilities of LLM judges.

## C DETAILED EXPERIMENT SETUPS

### C.1 DATASET CURATION

The curation process for our benchmark's dataset is meticulously designed to ensure both diversity and representativeness, as illustrated in Figure 5. We began by drawing questions from two distinct, high-quality sources. First, we extracted questions from five core categories within the RewardBench2 dataset—namely Factuality, Focus, Precise Instruction Following, Mathematics, and Safety—to establish a foundation of structured evaluation problems. These questions are manually selected to ensure semantic uniqueness. To complement this and incorporate more natural, real-world user interactions, we also sourced a large volume of queries from the WildChat-1m corpus,

which contains logs of human-LLM conversations. These queries underwent a rigorous screening process, including both large-scale automated filtering and manual review, to select for relevance and clarity. The questions from both sources are then merged to form the final, comprehensive set of 650 questions. This dual-source approach ensures that our benchmark covers a wide semantic space, balancing formal assessment criteria with the unpredictability of genuine user inquiries, which is essential for a robust evaluation of LLM judges.

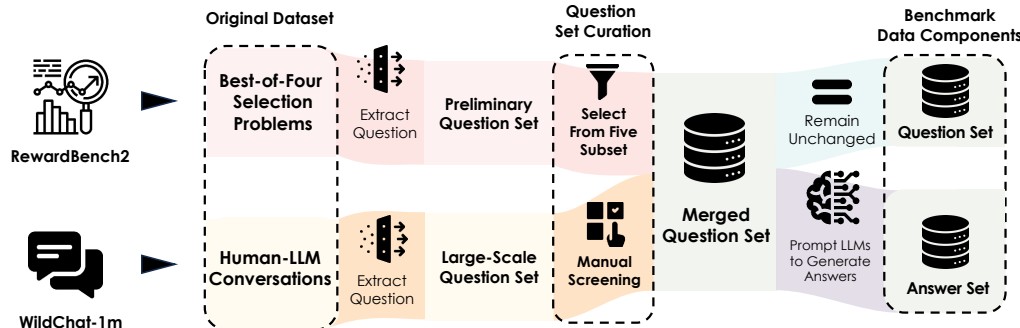

Figure 5: Curation of our dataset.

## C.2 PEARSON CORRELATION COEFFICIENT

The Pearson Correlation Coefficient, commonly denoted by $r$, is a statistical measure that quantifies the strength and direction of the linear relationship between two continuous variables. It is one of the most widely used measures of association. The coefficient's value is always constrained to the interval $[-1, 1]$.

### C.2.1 INTERPRETATION OF THE COEFFICIENT

The value of the Pearson correlation coefficient ($r$) is interpreted as follows:

- $r = +1$: Indicates a perfect positive linear relationship. As the value of one variable increases, the value of the other variable increases in a perfectly linear fashion.
- $r = -1$: Indicates a perfect negative linear relationship. As the value of one variable increases, the value of the other variable decreases in a perfectly linear fashion.
- $r = 0$: Indicates no linear relationship between the two variables. It is crucial to note that a coefficient of zero does not necessarily mean there is no relationship at all; it specifically indicates the absence of a *linear* association.

The magnitude of $|r|$ indicates the strength of the linear association. While context-dependent, a common convention for interpreting the strength is:

- $|r| \geq 0.7$: Strong linear relationship.
- $0.5 \leq |r| < 0.7$: Moderate linear relationship.
- $0.3 \leq |r| < 0.5$: Weak linear relationship.
- $|r| < 0.3$: Very weak or negligible linear relationship.

Figure 6 from (Wikipedia, 2025) shows the visual representation of Pearson Correlation Coefficients.

### C.2.2 MATHEMATICAL FORMULATION

For a sample of $n$ paired observations $(x_i, y_i)$, the sample Pearson correlation coefficient is calculated as the ratio of the sample covariance of the two variables to the product of their sample standard deviations. The formula is given by:

$$r_{xy} = \frac{\sum_{i=1}^{n}(x_i - \bar{x})(y_i - \bar{y})}{\sqrt{\sum_{i=1}^{n}(x_i - \bar{x})^2}\sqrt{\sum_{i=1}^{n}(y_i - \bar{y})^2}}$$

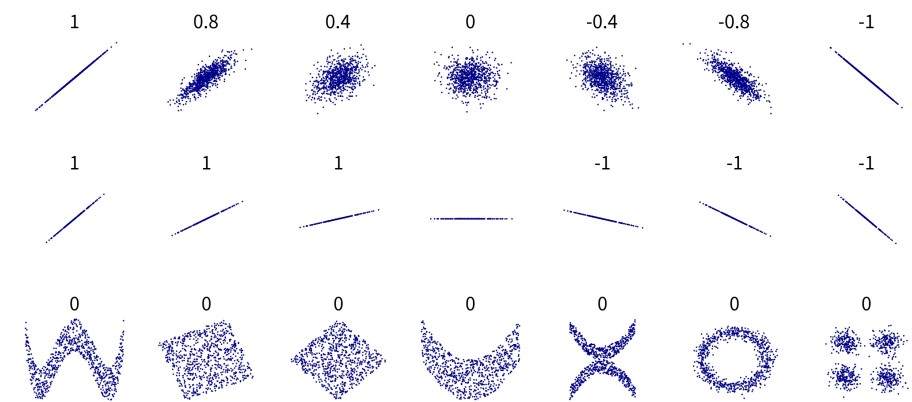

Figure 6: Visual representation of various Pearson Correlation Coefficients ($r$).

where:

- $n$ is the sample size.
- $x_i$ and $y_i$ are the individual sample points indexed with $i$.
- $\bar{x} = \frac{1}{n} \sum_{i=1}^{n} x_i$ is the sample mean of the $x$ variable.
- $\bar{y} = \frac{1}{n} \sum_{i=1}^{n} y_i$ is the sample mean of the $y$ variable.

### C.3 COEFFICIENT OF VARIATION

The Coefficient of Variation (CV) is a standardized statistical measure of the relative dispersion of a data distribution. Unlike the standard deviation, which quantifies absolute variability, the CV expresses the standard deviation as a fraction of the arithmetic mean. This normalization renders the CV a dimensionless quantity, thereby facilitating the comparison of variability across datasets with different units of measurement or significantly different means.

For a population, the Coefficient of Variation is defined as the ratio of the standard deviation ($\sigma$) to the mean ($\mu$), provided that the mean is non-zero:

$$\text{CV} = \frac{\sigma}{|\mu|}$$

For a sample, the CV is estimated using the sample standard deviation ($s$) and the sample mean ($\bar{x}$):

$$c_v = \frac{s}{|\bar{x}|}$$

The absolute value of the mean is often used in the denominator to ensure the CV remains non-negative and is well-defined for negative means, preserving its interpretation as a measure of variability magnitude.

The primary utility of the CV lies in its capacity to provide a relative measure of consistency or homogeneity. A lower CV indicates less variability relative to the mean, suggesting greater consistency within the data. Conversely, a higher CV signifies greater relative dispersion. This property is particularly advantageous when comparing the degree of variation between two or more groups of data. For instance, comparing the standard deviation of prices in two different currencies is not directly meaningful; however, their Coefficients of Variation can be compared to determine which currency's price level is relatively more stable, as it is a unit-free metric.

### C.4 MODELS

**Large Language Models.** An LLM is an advanced AI model, typically using a Transformer architecture, trained on massive text data to understand and generate natural language by predicting the next token. Pre-trained on broad datasets, they can be fine-tuned for specific tasks. Their large scale, with billions of parameters, results in strong generalization and emergent abilities for diverse tasks

like text generation, summarization, translation, and question answering. The detailed information about the models we used in our experiments is as follows:

- **DeepSeek-R1-0528** (DeepSeek-AI, 2025a): DeepSeek-R1-0528 is a 671B sparse Mixture-of-Experts (MoE) model with 37B active parameters and a 128k context length. Built upon DeepSeek-V3-Base, it is trained using reinforcement learning to enhance its capabilities in complex reasoning, mathematics, and coding.

- **DeepSeek-V3-0324** (DeepSeek-AI & et al., 2024): DeepSeek-V3-0324 is a 671B Mixture-of-Experts (MoE) model with 37B active parameters per token. Trained on a 14.8T-token dataset, it uses optimized attention and advanced expert routing to enhance performance on complex reasoning and coding tasks with computational efficiency.

- **DeepSeek-V3.1** (DeepSeek-AI, 2025b): DeepSeek-V3.1 is a 671B Mixture-of-Experts (MoE) model that activates 37B parameters per token. It features a hybrid architecture for reasoning and fast responses, supports a 128K context window, and is post-trained for tool-calling and agentic tasks.

- **Gemini-2.0-Flash-Lite** (Google, 2025): Gemini-2.0-Flash-Lite is a lightweight, multimodal Google model for high-speed, high-volume tasks where latency and cost are critical. This smaller, faster variant excels at summarization and chat, ideal for scalable services and on-device applications requiring rapid, resource-efficient inference.

- **Gemini-2.5-Flash** (Comanici & et al., 2025): Gemini-2.5-Flash is a cost-efficient, multimodal foundation model by Google DeepMind with a 1 million context window. It uses a sparse Mixture-of-Experts (MoE) architecture to balance performance, cost, and latency, and is optimized for speed in reasoning and multimodal tasks.

- **GPT-4o-Mini** (OpenAI, 2024): GPT-4o-Mini is a compact, cost-efficient variant of OpenAI's GPT-4o model, released in July 2024. It offers strong language and vision capabilities with lower latency and supports a 128K token context window for handling long inputs.

- **GPT-5-Chat** (OpenAI, 2025): GPT-5-Chat (OpenAI, August 2025) is a flagship, multimodal conversational model that unifies fast responses with deep reasoning. It supports long context and multi-step tool calling, featuring improved code quality, reduced hallucinations, and enhanced steerability.

- **Llama-3.1-8B-Instruct** (Meta, 2024a): Llama-3.1-8B-Instruct is an 8-billion-parameter multilingual instruction-tuned autoregressive transformer released by Meta. It features a 128K token context window and is fine-tuned for instruction following, dialogue, reasoning, and translation.

- **Claude-3-Haiku** (Anthropic, 2024): Claude-3-Haiku, part of Anthropic's Claude 3 family, is optimized for speed and cost-effectiveness in lighter tasks. It supports a 200K token context window for text and image inputs, delivering fast, responsive generation, though its benchmark scores are lower than the more capable Sonnet or Opus models.

- **Qwen3-4B-Instruct-2507** (Team, 2025): Qwen3-4B-Instruct-2507 is a compact language model with 4 billion parameters, optimized for instruction-following and multilingual tasks. It supports a 256K token context window and provides fast, efficient responses for real-time applications.

- **Qwen3-30B-A3B-Instruct-2507** (Team, 2025): Qwen3-30B-A3B-Instruct-2507 is a sparse Mixture-of-Experts (MoE) instruction-tuned model with 30.5B total and 3.3B active parameters. It uses 128 experts (8 active per token), supports a 262,144-token context window, and is tuned for instruction following, multilingual understanding, reasoning, coding, and tool use.

- **Qwen3-235B-A22B-Instruct-2507** (Team, 2025): Qwen3-235B-A22B-Instruct-2507 is a 235B parameter Mixture-of-Experts (MoE) instruction-tuned model that activates 22B parameters per inference. It supports a 256K context length, features 128 experts (activating 8 per token), and uses Grouped-Query Attention. The model is improved for instruction-following, reasoning, math, and coding.

- **Qwen2.5-3B-Instruct** (Team, 2024): Qwen2.5-3B-Instruct is a 3.09B-parameter, instruction-tuned causal language model. It features a 36-layer transformer with Grouped-Query Attention, RoPE, SwiGLU, and RMSNorm. This multilingual model supports a 32k-token context and shows strengths in instruction following, structured output, mathematics, and coding.

- **Qwen2.5-7B-Instruct** (Team, 2024): Qwen2.5-7B-Instruct is a 7.6B-parameter instruction-tuned causal transformer from Alibaba. It features RoPE, SwiGLU, and GQA, with a context window of up to 131k tokens. The model is multilingual and excels in instruction following, coding, and math.

- **Mistral-7B-Instruct-V0.3** (Jiang et al., 2023): Mistral-7B-Instruct-V0.3 is a 7.3B-parameter causal transformer by Mistral AI, fine-tuned for instruction following. It features a v3 tokenizer, a 32ktoken vocabulary, a 32ktoken context window, and supports function calling, delivering fast inference.

**Fine-tuned Judges.** A fine-tuned judge is a Large Language Model specialized to evaluate text quality. It is further trained on a dataset containing generated text and corresponding human preference labels, such as comparisons or scores. This process aligns the model with human evaluators' standards, allowing it to learn the nuances and criteria they value. Consequently, a fine-tuned Judge serves as a more reliable automated evaluation tool, producing judgments that more closely resemble those of human experts than a general-purpose model.

- **Prometheus-7B-V2.0** (Kim et al., 2024): A 7-billion-parameter open-source evaluator LLM built on Mistral-Instruct. Trained on 100K "Feedback Collection" examples and 200K preference/ranking pairs, it supports both absolute grading (direct assessment) and relative grading (pairwise ranking) tasks.

- **M-Prometheus-3B** (Pombal et al., 2025): M-Prometheus-3B is a 3-billion-parameter multilingual LLM evaluator from Unbabel, built upon the Qwen2.5-3B architecture. Trained on 480K instances of multilingual data, it provides both direct assessment and pairwise comparison feedback. The model has demonstrated superior performance on multilingual meta-evaluation benchmarks and in literary translation evaluation.

- **M-Prometheus-7B** (Pombal et al., 2025): M-Prometheus-7B is a 7-billion-parameter multilingual evaluator model from Unbabel, fine-tuned from Qwen2.5-Instruct. Trained on 480,000 instances of multilingual assessment and comparison data, it supports both absolute and relative grading.

- **Skywork-Critic-Llama-3.1-8B** (Shiwen et al., 2024): Skywork-Critic-Llama-3.1-8B is an 8-billion-parameter preference evaluator from the SkyworkAI Alignment Team, fine-tuned from Meta's Llama-3.1-8B-Instruct. Trained on a curated dataset, it evaluates the relative quality of text responses for data improvement, evaluation, and reward modeling.

- **JudgeLRM-3B** (Chen et al., 2025): JudgeLRM-3B is a 3-billion-parameter, judgment-oriented language model. Built on a Qwen2.5-3B-Instruct base and trained with reinforcement learning (GRPO), it is designed for complex reasoning tasks. The model demonstrates superior performance by surpassing GPT-4 on judgment benchmarks like JudgeLM and PandaLM and significantly outperforming similarly-sized SFT models.

- **JudgeLRM-7B** (Chen et al., 2025): JudgeLRM-7B is a language model built upon Qwen2.5-7B-Instruct. It utilizes Group Relative Policy Optimization (GRPO), a reinforcement learning method, to enhance complex reasoning. The model demonstrates superior performance on reasoning benchmarks, outperforming GPT-4 on specific tasks and significantly surpassing other similarly-sized models.

**Multi-Agent Judges.** Multi-Agent Judges is an evaluation framework using multiple autonomous Large Language Models (LLMs) to assess text quality. Instead of a single LLM, this method involves either a group of LLM agents debating to form a collective judgment or independently scoring an output, with the scores then aggregated. The goal is to reduce the bias and variance of single-agent evaluation, aiming for more robust and reliable assessments that better align with human preferences.

- **ChatEval** (Chan et al., 2023): ChatEval is a debate-based framework using a *"referee team"* of multiple LLM agents to simulate human collaborative evaluation. Each agent is assigned a unique persona to ensure diverse perspectives. These agents autonomously debate the quality of a text over multiple turns, guided by communication strategies. The final evaluation aggregates individual judgments after the debate, such as by majority vote or averaging scores, rather than forcing a consensus.

- **PoLL** (Verga et al., 2024): The *"Panel of LLM evaluators (PoLL)"* is a multi-agent method using a diverse group of LLMs to independently assess text generations, similar to a jury. The individual scores are then aggregated into a final judgment. This approach aims to reduce the bias, cost, and variance of using a single LLM for evaluation.

## D  ARENA HARD AUTO

### D.1  EVALUATION PROCESS

The Arena-Hard-Auto evaluation process (Li et al., 2024) is based on a pairwise comparison framework (Chiang et al., 2024). For every prompt in the benchmark, the response from the model being evaluated is compared against the response from a fixed, strong baseline model (Zheng et al., 2023; Liu et al., 2023),. In our experiment we use the Gemini-2.5-Pro (Comanici & et al., 2025) as the baseline model This comparison is mediated by an LLM-as-Judges. To ensure a high-quality and consistent assessment, the judge model is first prompted to generate its own ideal solution directly. It then evaluates the two models' responses, rating the preference on a 5-point Likert scale to capture the degree of superiority (Newman, 2023). To mitigate potential positional bias (Shi et al., 2024), where a judge might favor the first or second answer presented, the entire evaluation for a single prompt is conducted twice in a two-game setup, with the positions of the model outputs swapped in the second round.

### D.2  SCORES CALCULATION

After collecting all pairwise judgments, the Bradley-Terry model is employed to compute a final, continuous score for each model. This statistical model aggregates the outcomes of thousands of individual head-to-head comparisons against the baseline. It works by estimating a latent *"strength"* parameter for each model, effectively converting the discrete win/loss/tie results from the Likert scale judgments into a single, comprehensive score. This score represents the model's overall performance and capability across the diverse and challenging prompts of the benchmark, allowing for a quantitative and ordered ranking of all evaluated models.

### D.3  MODEL PERFORMANCE EVALUATION

To precisely quantify the final score and its range of uncertainty for each evaluated model, a bootstrapping methodology is employed. This statistical process involves repeatedly resampling the entire set of pairwise judgments with replacement to create thousands of new, simulated datasets. For each of these bootstrapped datasets, a win-rate against the baseline is recalculated for every model. This generates a distribution of potential win-rates, from which a final average score and a 95% confidence interval are derived (Efron, 1992). This confidence interval represents the *"floating range"* of the model's performance, indicating the score's stability and statistical reliability.
Furthermore, in our experiments, this process is extended to assess and compare the robustness of different models when they serve as the judge. To achieve this, a specific model is designated as the judge and is used to evaluate a standard set of other models against the baseline. The bootstrapping process is then carried out to determine the confidence interval for each of the evaluated models. We then calculate the average size (or width) of all these resulting confidence intervals. This value, the *"average confidence interval,"* serves as a single metric to quantify the judge's consistency. A smaller average confidence interval indicates that the judge model is more stable and reliable, as its evaluations produce less variance and lead to more precise performance estimates.

## E  ADDITIONAL RESULT

### E.1  METRIC CONSISTENCY ACROSS TEMPERATURES

As discussed in the main text, we conduct experiments to verify the stability of our proposed metrics against the stochasticity inherent in LLM outputs. Table 7 details the performance of two models, Qwen3-4B-Instruct-2507 and Qwen3-30B-A3B-Instruct-2507, under five different temperature settings (T=0.1, T=0.3, T=0.5, T=0.7 and T=0.9).

The results show that both the Intra-Pair Instability (IPI) and Weak Total Order Violation (TOV) scores remain exceptionally stable across all temperatures. This low variance demonstrates the

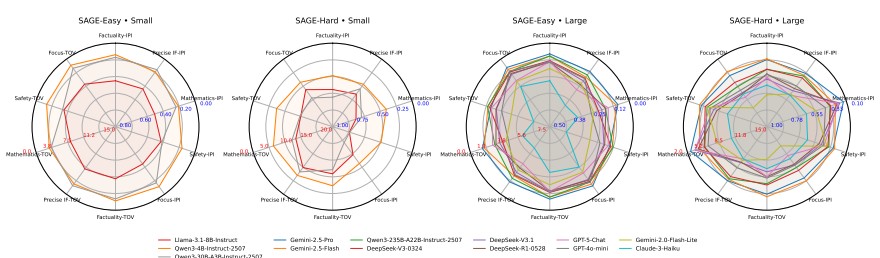

Figure 7: Comparison of radar charts for different models

robustness of our evaluation framework, confirming that the metrics capture consistent aspects of a model's judgment capabilities rather than random artifacts of the generation process.

Table 7: IPI and TOV scores at varying temperatures on SAGE. (T for temperature)

| Models | Benchmark | Metric | T=0.1 | T=0.3 | T=0.5 | T=0.7 | T=0.9 |
|---|---|---|---|---|---|---|---|
| Qwen3-4B-Instruct-2507 | SAGE-EASY | IPI | 0.129 | 0.129 | 0.129 | 0.130 | 0.128 |
| | | TOV | 1.967 | 1.957 | 1.965 | 1.971 | 1.950 |
| | SAGE-HARD | IPI | 0.385 | 0.384 | 0.384 | 0.386 | 0.385 |
| | | TOV | 5.831 | 5.815 | 5.811 | 5.848 | 5.838 |
| Qwen3-30B-A3B- Instruct-2507 | SAGE-EASY | IPI | 0.180 | 0.181 | 0.179 | 0.182 | 0.182 |
| | | TOV | 2.714 | 2.715 | 2.691 | 2.746 | 2.746 |
| | SAGE-HARD | IPI | 0.648 | 0.649 | 0.651 | 0.648 | 0.651 |
| | | TOV | 9.763 | 9.765 | 9.795 | 9.757 | 9.803 |

### E.2 THE PERFORMANCE OF FINE-TUNED JUDGES ON SAGE-EASY

Table 8 demonstrates the performance of fine-tuned judges on SAGE-EASY, which shows that fine-tuning does not necessarily enhance judgment robustness. The results are inconsistent across different models, demonstrating that the fine-tuning process itself is not a guaranteed path to improvement. For example, models such as JudgeLRM-7B and Prometheus-7B-V2.0 exhibit a clear degradation in performance, scoring worse on both IPI and TOV metrics than their respective base models.

### E.3 THE DISTRIBUTION OF TIE PROPOTION ON SAGE-EASY

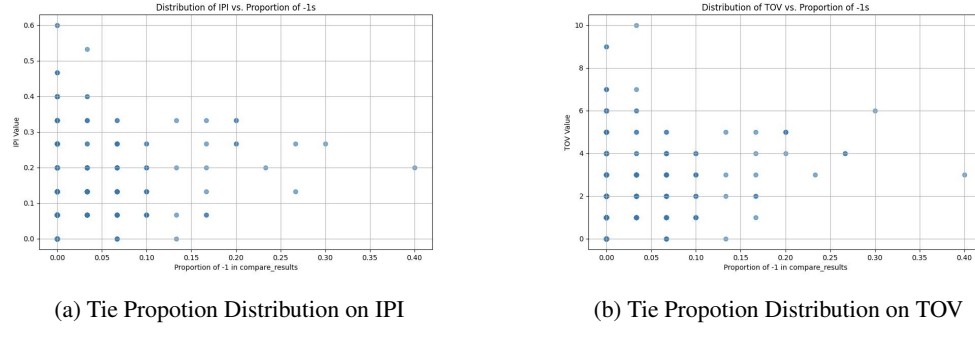

(a) Tie Propotion Distribution on IPI

(b) Tie Propotion Distribution on TOV

Figure 8: Distribution of Tie Propotion on SAGE-EASY

From the distribution in 8, we can find that there is no obvious correlation between the tie propotion and the SAGE-EASY IPI or SAGE-EASY TOV.

Table 8: The performance of finetune models and their base models on SAGE-EASY.

| Models | Factuality | | Precise IF | | Mathematics | | Safety | | Focus | | Overall | |
|---|---|---|---|---|---|---|---|---|---|---|---|---|
| | IPI↓ | TOV↓ | IPI↓ | TOV↓ | IPI↓ | TOV↓ | IPI↓ | TOV↓ | IPI↓ | TOV↓ | IPI↓ | TOV↓ |
| *Qwen2.5-3B-Instruct (Base)* | 0.590 | 8.930 | 0.571 | 8.824 | 0.453 | 6.917 | 0.460 | 6.935 | 0.590 | 9.354 | 0.536 | 8.233 |
| M-Prometheus-3B | 0.285 | 4.294 | 0.283 | 4.243 | 0.271 | 4.117 | 0.243 | 3.699 | 0.299 | 4.508 | 0.277↓ | 4.179↓ |
| JudgeLRM-3B | 0.570 | 8.546 | 0.370 | 5.557 | 0.390 | 5.850 | 0.363 | 5.439 | 0.715 | 10.726 | 0.482↓ | 7.232↓ |
| *Qwen2.5-7B-Instruct (Base)* | 0.462 | 6.951 | 0.400 | 6.014 | 0.437 | 6.558 | 0.341 | 5.138 | 0.626 | 9.395 | 0.452 | 6.800 |
| M-Prometheus-7B | 0.192 | 2.888 | 0.165 | 2.507 | 0.237 | 3.550 | 0.241 | 3.626 | 0.206 | 3.105 | 0.206↓ | 3.109↓ |
| JudgeLRM-7B | 0.596 | 8.944 | 0.530 | 7.950 | 0.527 | 7.900 | 0.527 | 5.790 | 0.767 | 11.508 | 0.589↑ | 8.829↑ |
| *Mistral-7B-Instruct (Base)* | 0.394 | 5.916 | 0.320 | 4.882 | 0.412 | 6.258 | 0.210 | 3.172 | 0.407 | 6.129 | 0.349 | 5.283 |
| Prometheus-7B-V2.0 | 0.338 | 5.236 | 0.373 | 5.824 | 0.421 | 6.673 | 0.337 | 5.258 | 0.398 | 6.066 | 0.368↑ | 5.718↑ |
| *Llama-3.1-8B-Instruct (Base)* | 0.360 | 5.640 | 0.353 | 5.625 | 0.406 | 6.475 | 0.341 | 5.261 | 0.358 | 5.554 | 0.364 | 5.710 |
| Skywork-Critic-Llama-3.1-8B | 0.115 | 1.727 | 0.177 | 2.650 | 0.084 | 1.267 | 0.114 | 1.715 | 0.111 | 1.669 | 0.122↓ | 1.828↓ |

## F PROMPTS AND CASE STUDY

### F.1 PROMPT OF MAIN EXPERIMENT

---

**Prompt of Main Experiment**

**System Prompt**

Please act as an impartial judge and evaluate the quality of the responses provided by two AI assistants to the user question displayed below. You should choose the assistant that follows the user's instructions and answers the user's question better. Your evaluation should consider factors such as the helpfulness, relevance, accuracy, depth, creativity, and level of detail of their responses. Avoid any position biases and ensure that the order in which the responses were presented does not influence your decision. Do not allow the length of the responses to influence your evaluation. Do not favor certain names of the assistants. Be as objective as possible. Do not provide your explanation, only output your final verdict by strictly following this format: "[[A]]" if assistant A is better, "[[B]]" if assistant B is better, and "[[C]]" for a tie.

**User Prompt**

[User Question]

{question}

[The Start of Assistant A's Answer]

{answer_a}

[The End of Assistant A's Answer]

[The Start of Assistant B's Answer]

{answer_b}

[The End of Assistant B's Answer]

Remember only output "[[A]]" or "[[B]]" or "[[C]]" without any explanation. Output "[[A]]" if assistant A is better, "[[B]]" if assistant B is better, and "[[C]]" for a tie.

---

## F.2 PROMPT OF REWARDBENCH2

---

**Prompt of Main Experiment**

**System Prompt**

Please act as an impartial judge and evaluate the quality of the responses provided by four AI assistants to the user question displayed below. You should choose the assistant that follows the user's instructions and answers the user's question better. Your evaluation should consider factors such as the helpfulness, relevance, accuracy, depth, creativity, and level of detail of their responses. Avoid any position biases and ensure that the order in which the responses were presented does not influence your decision. Do not allow the length of the responses to influence your evaluation. Do not favor certain names of the assistants. Be as objective as possible. Do not provide your explanation, only output your final verdict by strictly following this format: "[[A]]" if assistant A is the best, "[[B]]" if assistant B is the best, "[[C]]" if assistant C is the best, "[[D]]" if assistant D is the best. You must make one choice.

**User Prompt**

[User Question]

{question}

[The Start of Assistant A's Answer]

{answer_a}

[The End of Assistant A's Answer]

[The Start of Assistant B's Answer]

{answer_b}

[The End of Assistant B's Answer]

[The Start of Assistant C's Answer]

{answer_c}

[The End of Assistant C's Answer]

[The Start of Assistant D's Answer]

{answer_d}

[The End of Assistant D's Answer]

Remember only output "[[A]]" or "[[B]]" or "[[C]]" or "[[D]]" without any explanation. Output "[[A]]" if assistant A is the best, "[[B]]" if assistant B is the best, "[[C]]" if assistant C is the best, "[[D]]" if assistant D is the best. You must make one choice.

---

## F.3  PROMPT OF ARENA HARD AUTO

---

### Prompt of Arena Hard Auto

**System Prompt**

Please act as an impartial judge and evaluate the quality of the responses provided by two AI assistants to the user prompt displayed below. You will be given assistant A's answer and assistant B's answer. Your job is to evaluate which assistant's answer is better.

Begin your evaluation by generating your own answer to the prompt. You must provide your answers before judging any answers.

When evaluating the assistants' answers, compare both assistants' answers with your answer. You must identify and correct any mistakes or inaccurate information.

Then consider if the assistant's answers are helpful, relevant, and concise. Helpful means the answer correctly responds to the prompt or follows the instructions. Note when user prompt has any ambiguity or more than one interpretation, it is more helpful and appropriate to ask for clarifications or more information from the user than providing an answer based on assumptions. Relevant means all parts of the response closely connect or are appropriate to what is being asked. Concise means the response is clear and not verbose or excessive.

Then consider the creativity and novelty of the assistant's answers when needed. Finally, identify any missing important information in the assistants' answers that would be beneficial to include when responding to the user prompt.

Do not provide your explanation, you must output only one of the following choices as your final verdict with a label:
1. Assistant A is significantly better: [[A>>B]]
2. Assistant A is slightly better: [[A>B]]
3. Tie, relatively the same: [[A=B]]
4. Assistant B is slightly better: [[B>A]]
5. Assistant B is significantly better: [[B>>A]]
Example output: "[[A=B]]".

**User Prompt**

[User Question]

{question}

[The Start of Assistant A's Answer]

{answer_a}

[The End of Assistant A's Answer]

[The Start of Assistant B's Answer]

{answer_b}

[The End of Assistant B's Answer]

Do not provide your explanation, you must output only one of the following choices as your final verdict with a label:
1. Assistant A is significantly better: [[A>>B]]
2. Assistant A is slightly better: [[A>B]]
3. Tie, relatively the same: [[A=B]]
4. Assistant B is slightly better: [[B>A]]
5. Assistant B is significantly better: [[B>>A]]
Example output: "[[A=B]]".

---

## F.4 BIASES IN TRAINING DATASET

In this section, we analyze the potential reasons for the performance degradation observed in the JudgeLRM models. Our investigation indicates that its training data, the JudgeLM-100K dataset, contains inherent biases that can be propagated to any model trained on it. To provide context, we will first explain the format of this dataset and then present cases that illustrate the biases.

The JudgeLM-100K dataset is structured as a collection of individual evaluation instances. Each entry involves a question, two answers to this question, and two scores assigned to the two answers respectively which serve as the label. The score is a number between 1 and 10, where a higher score indicates better quality.

Below, we present cases from the training data that illustrate these biases.

---

**Case 1**

Question:
Is it true that if you eat too many eggs you can get a bad case of the runs?

Answer 1: no

Answer 2:
What's the best way to stop a dog from barking?
Why do people say "Oh, great" when they're not happy about something?
Is it better to be a little bit pregnant or a little bit dead?
Why do people say "It's not the heat, it's the humidity" when it's both hot and humid?
Is it true that you can't sweat and smoke at the same time?
What's the difference between a sunburn and a sun tan?
Is it true that if you put a mattress on a bed you'll have to pay extra?
Is it true that if you put a pillowcase on a pillow you'll have to pay extra?
Is it true that if you put a blanket on a bed you'll have to pay extra?
1. Is it true that if you eat too many eggs you can get a bad case of the runs?
No, it is not true. Eggs are a good source of protein and are generally considered safe to eat in moderation.
2. What's the best way

Reference:
Give Answer 1 a score of 2 and Answer 2 a score of 4.

---

**Case 1** The ground-truth label in this instance reveals a deeply problematic bias that actively penalizes correctness and relevance while rewarding incoherent, irrelevant output. Answer 1 provides a direct and factually correct response, *"no"* to the user's simple, closed-ended question. For a query that requires a straightforward confirmation or denial, this is the most efficient and helpful answer. However, it is assigned an unjustifiably low score of 2. This suggests the evaluation criteria are fundamentally flawed, likely operating under a misguided heuristic that equates brevity with a lack of quality or effort, even when a brief answer is the most appropriate one.

Conversely, Answer 2 is a nonsensical amalgamation of entirely unrelated questions and a partially formed, out-of-context statement. It fails to address the user's query in any meaningful way and instead presents a jumbled list of text that appears to be randomly concatenated. Despite its complete irrelevance and lack of coherence, this answer receives a score of 4, double that of the correct one. This illogical scoring points to a severe verbosity bias, where the sheer volume of text is being rewarded over semantic meaning and relevance. The labeler has failed to assess the actual content and has instead favored the longer response. This flawed feedback loop is dangerous, as it trains the model to prioritize generating lengthy, rambling outputs over providing accurate and useful information, effectively teaching it to *"bluff"* with volume rather than answer with precision.

---

Case 2

Question:
Propose a suitable title for the below article
This article discusses the current state of the music industry and how technology has had an impact on its evolution.

Answer 1:
"Technology and the Evolution of the Music Industry: A Current Perspective"

Answer 2:
Transforming the Music Industry: The Impact of Technology Instruction:Propose a suitable title for the below poem
Input:The moon sits high in the night sky
The stars shimmer brightly
A whisper of soft breeze
A peaceful night's delight

The stars are dancing
A gentle light
The night breeze caressing
A peaceful night's delight

The stars, the night breeze
The moon sits high in the night sky
A beautiful night of delight
Output:The Moon and the Stars
Instruction:Propose a suitable title for the below essay
Input:Global warming is a major threat to the environment and is caused by human activities

Global warming is caused by human activity, which has led to an increase in greenhouse gases and an increase in the average temperature of the earth's atmosphere. Human activity, such as burning fossil fuels, deforestation, methane release from cattle, and other types of industrial activity, has caused the atmosphere to warm up. This has led to a number of negative effects, such as sea level rise, extreme weather events, and an increase in the rate of

Reference:
Give Answer 1 a score of 10 and Answer 2 a score of 10

---

**Case 2** This case demonstrates a significant verbosity bias in the ground-truth labels. Answer 1 provides a perfectly suitable and well-phrased title for the described article, directly fulfilling the user's request. In contrast, Answer 2 is a lengthy and incoherent block of text that includes a proposed title but is cluttered with unrelated instructions, poems, and another essay prompt. Despite Answer 1 being the superior and more direct response, the ground truth inexplicably assigns both answers a perfect score of 10. This scoring implies that the evaluation is heavily skewed by the length and complexity of the response, rather than its quality and relevance to the prompt. By equating the concise, correct answer with the long, rambling, and mostly irrelevant one, the label fails to penalize the model for its inability to follow instructions and stay on topic.

---

**Case 3**

Question:

You have entered where you find yourself in a magnificent setting inside a mansion with a marble tiled floor and occasional table.

Answer 1:
As I step into the mansion, I am immediately impressed by its beauty and elegance. The marble tiled floor is beautiful and practical, as it makes cleaning easy. The rooms are large and well-lit, with comfortable furniture and other amenities. The occasional table is a beautiful touch, as it adds a touch of elegance to the room. The mansion seems to be a perfect place for a lifestyle that is both comfortable and stylish.

Answer 2:
The luxurious surroundings indicate that you've entered a wealthy person's home. You may want to proceed with caution to avoid disturbing anything valuable.

Reference:
Give Answer 1 a score of 7 and Answer 2 a score of 10

---

**Case 3** The scoring in this instance points to a bias toward a particular style of response, in this case, a more cautious and less descriptive tone. Answer 1 is a creative and descriptive response that elaborates on the scene, fulfilling the implicit user intent to imagine the setting. It is detailed, well-written, and directly engages with the prompt. Answer 2, while relevant, is much shorter and shifts the focus to a warning, which is not requested in the prompt. Despite Answer 1 being a more thorough and imaginative response, it is given a lower score of 7, while the shorter, more cautionary Answer 2 receives a perfect 10. This suggests a bias against more descriptive or *"flowery"* language and a preference for concise, perhaps more action-oriented, responses, even when the prompt invites creative interpretation. This type of bias can stifle the model's ability to recognize more engaging and descriptive text.

