# OpenReview forum: "Sage: A Scalable Framework for Evaluating LLM-as-a-Judge Without Human Effort"
_ICLR.cc/2026/Conference — Submitted to ICLR 2026_

### Official Review · Reviewer_1XJN · 2025-10-26

**Soundness:** 2
**Presentation:** 3
**Contribution:** 2
**Rating:** 4
**Confidence:** 3

**Summary:**

SAGE proposes a fully automatic framework to evaluate “LLM-as-a-Judge” without human gold labels by measuring (i) local self-consistency via a symmetrized pairwise protocol (Intra-Pair Instability, IPI) and (ii) global logical coherence via the minimum edits to reach a weak total order (TOV). Using 650 questions (RewardBench2 + WildChat-1M) and 6 candidate answers per question, the paper reports low metric variance, strong correlations with existing human-grounded judge benchmarks, and analyses showing that rubric-guided judging and multi-model panels help, while fine-tuned judges can regress.

**Strengths:**

- Clear, axiomatic motivation for order-consistency as a desideratum for LLM judges; precise formalization of IPI (positional self-consistency) and TOV (global transitivity to a weak total order).
- Practical, scalable evaluation protocol (round-robin, symmetrized) with easy-to-reproduce ingredients; sensible EASY vs HARD tiers via answer diversity vs homogeneity.
- Empirical metric stability (very low variance) and reported correlations with LLMBar/RewardBench2 that argue SAGE is at least a useful proxy for robustness/accuracy without human labels.
- Useful analyses: rubric prompting reduces IPI/TOV, model “reasoning depth” helps, mixed outcomes for fine-tuned judges, and modest gains from panel aggregation.
- Ethical/reproducibility commitments (release of code/dataset/prompts; curated sourcing).

**Weaknesses:**

- Proxy vs validity gap: IPI/TOV assess internal logical/positional consistency, not external correctness; the paper relies on correlations to argue usefulness, but this does not establish causal or sufficient validity—especially on subjective tasks. More direct evidence is needed that lowering IPI/TOV truly improves application outcomes (e.g., arena rankings stability, RLHF learning signals).
- Model- and prompt-sensitivity: The symmetrized protocol and prompts may themselves induce artifacts (e.g., sensitivity to tie handling, abstentions, verbosity formatting). The paper does not systematically vary prompt templates, tie policies, or judge “instruction strength” and report robustness envelopes.
- TOV computation & scalability: Minimizing edits to a weak total order is a nontrivial combinatorial problem; complexity/algorithmic details and performance for larger n are not fully fleshed out (the work uses n=6). Practicality for n≫6 and the effect on confidence intervals are unclear.
- Construction of HARD set: “Hardness” is instantiated by generating all 6 answers from a single capable model. This conflates source homogeneity with true near-ties, and risks model-specific quirks. A more model-agnostic construction or human sanity checks on difficulty would strengthen claims.
- Multi-agent analysis is thin: The negative ChatEval result may be configuration-dependent; ablations (num-rounds, argument exchange format, judge selection) are missing, so the conclusion that “debate fails to help” is too strong.
- No cost/throughput accounting: A selling point is scalability “without human effort,” but the token/runtime cost of full round-robin, symmetrized judging across 650×C(6,2) pairs and multiple runs is not reported; practicality for frequent benchmarking or larger suites is unclear.
- External impact evidence: Claims that SAGE improves arena stability and RLHF safety are suggestive; stronger end-to-end studies (e.g., swap reward model/arena judge using SAGE-selected judges and show better downstream stability/accuracy) are missing.

**Questions:**

- Can you run an ablation where the only change in an arena/auto-rater or RLHF loop is replacing the judge with a SAGE-selected one, and show narrower Elo CIs or higher human agreement?
- Can you report results under ≥3 substantially different judge prompts (instruction strength, rubric placement, formatting) and show that rankings and absolute scores remain stable?
- How frequently does the judge return “equal quality”? How are frequent ties treated in IPI/TOV, and how sensitive are the metrics to the proportion of ties?

---

> ### Author Response · Authors · 2025-11-19
> **Response to Reviewer 1XJN (1)**
>
> Thank you very much for your valuable feedback. During this interval, we tried our best to address all your concerns and revised our paper based on your advice. We will address each of your concerns and provide explanations to help you better our latest update of this paper step by step:
>
> **Question 1:** Can you run an ablation where the only change in an arena/auto-rater or RLHF loop is replacing the judge with a SAGE-selected one, and show narrower Elo CIs or higher human agreement?
>
> **Answer:**  Thank you for the insightful suggestion. We agree that demonstrating the practical utility of SAGE in selecting better judges is crucial.
>
> We believe our current experiments in **Section 4.2** and **Section 4.4 (Page 9)** directly address this hypothesis, providing statistical evidence across 13 models rather than a single ablation case:
>
> 1. **Narrower Elo CIs:** In Section 4.4 ("Selecting Robust Judges for Automated Arenas"), we explicitly investigated the relationship between SAGE scores and the stability of Elo ratings in Arena-Hard-Auto. We found a strong positive correlation (Pearson Correlation Coefficient ≈ 0.76) between our metrics (IPI/TOV) and the **Elo rating confidence intervals (CIs)**.
>
>
>
> 2. **Higher Human Agreement:** In Section 4.2 and Table 3, we demonstrated that SAGE scores strongly correlate with performance on human-annotated benchmarks like **LLMBar** and **RewardBench2** (Pearson Correlation Coefficient up to 0.87). This confirms that a SAGE-selected judge serves as a highly effective proxy for human agreement and accuracy.
>
> To directly address the reviewer's suggestions for an ablation study, we compare two representative models from our experiments: **Qwen3-4B-Instruct** (a standard baseline) and **Gemini-2.5-Pro** (the optimal judge selected by our SAGE framework).
>
> As shown in the **Table 1**, replacing the baseline judge with the SAGE-selected judge results in a significant improvement across all dimensions.
>
> **Table 1: Sage-selected judge brings about higher human agreement and narrower ELO CIs.**
>
> | Feature                               | Baseline Judge        | Sage-Selected Judge | Improvement / Impact                        |
> | :------------------------------------ | :-------------------- | :------------------ | :------------------------------------------ |
> | **Model**                             | **Qwen3-4B-Instruct** | **Gemini-2.5-Pro**  | —                                           |
> | **Sage Metric: IPI**      | 0.129                 | **0.082**           | **36.4% Reduction** in local inconsistency  |
> | **Sage Metric: TOV**     | 1.952                 | **1.265**           | **35.2% Reduction** in global inconsistency |
> | **Human Agreement on Reward Bench 2** | 0.59                  | **0.75**            | **Higher Human Agreement**                  |
> | **Human Agreement on LLMBar**         | 0.60                  | **0.82**            | **Higher Human Agreement**                  |
> | **ELO CIs**                           | 0.037                 | **0.023**           | **Narrower ELO CIs**                        |
>
> ------

---

> ### Author Response · Authors · 2025-11-19
> **Response to Reviewer 1XJN (2)**
>
> **Question 2:** Can you report results under ≥3 substantially different judge prompts (instruction strength, rubric placement, formatting) and show that rankings and absolute scores remain stable?
>
> **Answer:** Thank you for this valuable suggestion. We fully agree with you. To address this concern, we conducted additional experiments using three substantially different prompt variations：
>
> **Additional Prompt1**:
> ```
> Act as a neutral evaluator and judge the quality of the two AI assistant responses below. Your evaluation should be based on the following criteria:
> - **Helpfulness & Relevance:** How well does the response address the user's core question?
> - **Accuracy & Depth:** Is the information correct and detailed?
> - **Clarity & Creativity:** Is the response hard to understand and well-structured?
>
> Disregard response length and presentation order to avoid bias. Conclude with a detailed explanation, followed by your final decision in this exact format: "[[A]]" if A is better, "[[B]]" if B is better, or "[[C]]" for a tie.
>
> [User Question]
> {question}
>
> [Assistant A's Answer]
> {answer_a}
>
> [Assistant B's Answer]
> {answer_b}
>
>
> Provide your analysis based on the specified criteria. Then, output your final verdict in the required format: "[[A]]", "[[B]]", or "[[C]]".
> ```
>
> **Additional Prompt2**:
>
> ```
> Please act as an impartial judge and evaluate the quality of the responses provided by two AI assistants to the user question displayed below. You should choose the assistant that follows the user's instructions and answers the user's question better. Your evaluation should consider factors such as the helpfulness, relevance, accuracy, depth, creativity, and level of detail of their responses. Avoid any position biases and ensure that the order in which the responses were presented does not influence your decision. Do not allow the length of the responses to influence your evaluation. Do not favor certain names of the assistants. Be as objective as possible. Do not provide your explanation, only output your final verdict by strictly following this format: "[[A]]" if assistant A is better, "[[B]]" if assistant B is better, and "[[C]]" for a tie.
>
> [User Question]
> {question}
>
> [The Start of Assistant A's Answer]
> {answer_a}
> [The End of Assistant A's Answer]
>
> [The Start of Assistant B's Answer]
> {answer_b}
> [The End of Assistant B's Answer]
>
> Remember only output "[[A]]" or "[[B]]" or "[[C]]" without any explanation. Output "[[A]]" if assistant A is better, "[[B]]" if assistant B is better, and "[[C]]" for a tie.
> ```
>
> **Additional Prompt 3**:
>
> ```
> Please serve as an objective arbiter. Your mission is to evaluate the two AI assistant responses provided below. Determine which assistant provides a better answer by considering its helpfulness, accuracy, relevance, and level of detail. It is crucial that your judgment is not influenced by the order of presentation or the length of the text. Be as impartial as possible. After your explanation, state your final decision using this precise format: "[[A]]" if Assistant A is better, "[[B]]" if Assistant B is better, or "[[C]]" for a tie.
>
> ### USER QUESTION ###
> {question}
>
>
> ### ASSISTANT A's ANSWER ###
> {answer_a}
>
>
> ### ASSISTANT B's ANSWER ###
> {answer_b}
>
>
> Provide your explanation, then conclude with your final verdict: "[[A]]", "[[B]]", or "[[C]]".
> ```
>
> We use these three prompts to test Qwen3-4B-Instruct-2507 on our benchmark. The results are shown in **Table 2**.
>
> **Table 2: Stability of SAGE metrics across different prompt variations. Easy-IPI means the IPI metric tested on Sage-Easy. Columns with the suffix **“-R”** (e.g., *Easy-IPI-R*) indicate the **ranking** of the models derived from the corresponding metric. The stability of rankings across different prompt styles demonstrates the reliability of our evaluation framework.**
>
> | Prompt              | Easy-IPI($\downarrow$) | Easy-IPI-R     | Easy-TOV($\downarrow$) | Easy-TOV-R     | Hard-IPI($\downarrow$) | Hard-IPI-R | Hard-TOV($\downarrow$) | Hard-TOV-R |
> | ------------------- | ---------------------- | -------------- | ---------------------- | -------------- | ---------------------- | ---------- | ---------------------- | ---------- |
> | Original Prompt     | 0.129                  | 4              | 1.952                  | 4              | 0.386                  | 5          | 5.849                  | 5          |
> | Additional Prompt 1 | 0.137 ($\uparrow$6%)   | 5($\uparrow$1) | 2.080($\uparrow$7%)    | 5($\uparrow$1) | 0.387($\uparrow$3‰)    | 5(=)       | 5.928($\uparrow$1%)    | 5(=)       |
> | Additional Prompt 2 | 0.126($\downarrow$2%)  | 4(=)           | 1.919($\downarrow$2%)  | 4(=)           | 0.388($\uparrow$1%)    | 5(=)       | 6.078($\uparrow$4%)    | 5(=)       |
> | Additional Prompt 3 | 0.132($\uparrow$2%)    | 5($\uparrow$1) | 2.025($\uparrow$4%)    | 5($\uparrow$1) | 0.399($\uparrow$3%)    | 5(=)       | 6.213($\uparrow$6%)    | 5(=)       |

---

> ### Author Response · Authors · 2025-11-19
> **Reponse to Reviewer 1XJN (3)**
>
> **Question 3:** How frequently does the judge return “equal quality”? How are frequent ties treated in IPI/TOV, and how sensitive are the metrics to the proportion of ties?
>
> **Answer:** We thank the reviewe for this insightful question regarding the prevalence of “tie” judgments and their influence on our metrics. We address your inquiry in three parts: the empirical frequency of ties, the methodological treatment of ties in our metrics, and a statistical sensitivity analysis.
>
> **1. Frequency of "Equal Quality" Judgments**
> We have calculated the proportion of “equal quality” (tie) outcomes returned by each of the 13 evaluated models across both Sage-Easy and Sage-Hard subsets. As shown in **Table 3**, the frequency of ties varies by model but remains within a reasonable range for most evaluators.
>
> **Table 3: The proportion of “equal quality” judgments returned by the 13 evaluated LLMs on Sage-Easy and Sage-Hard.**
>
> | Models        | Claude-3-Haiku | DeepSeek-R1 | DeepSeek-V3.1 | DeepSeek-V3_0324 | Gemini-2.0-Flash-Lite | Gemini-2.5-Flash | Gemini-2.5-Pro | GPT-4o-mini | GPT-5-Chat | Llama-3.1-8B-Instruct | Qwen3-4B | Qwen3-30B | Qwen3-235B-A22B |
> | ------------- | -------------- | ----------- | ------------- | ---------------- | --------------------- | ---------------- | -------------- | ----------- | ---------- | --------------------- | -------- | --------- | --------------- |
> | **SAGE-EASY** | 11.76%         | 2.62%       | 6.46%         | 1.75%            | 3.81%                 | 1.27%            | 1.83%          | 1.2%        | 1.28%      | 1.55%                 | 1.51%    | 1.36%     | 1.59%           |
> | **SAGE-HARD** | 37.8%          | 20.0%       | 41.9%         | 13.6%            | 17.4%                 | 12.3%            | 15.1%          | 12.0%       | 11.0%      | 5.99%                 | 13.3%    | 13.1%     | 15.3%           |
>
> **2. Treatment of Frequent Ties in IPI and TOV:**
> Our metrics are designed to handle ties as a valid preference state, provided they are consistent and transitive:
>
> - **In Intra-Pair Instability (IPI):** A tie is treated as a specific judgment class ($y_{ij}=0$). IPI measures local consistency under symmetry. A tie contributes to a low (better) IPI score *only if* $y_{ij}=0$ and $y_{ji} = 0$. If a model returns "equal quality" in the forward pass but "A is better" in the reverse pass, this is penalized as an inconsistency.
>
> - **In Weak Total Order Violation (TOV):** TOV calculates the minimum edits required to fit the judgments into a *weak total order*, which mathematically allows for indifference (ties). Therefore, a set of judgments containing ties is not inherently penalized. However, logical contradictions involving ties (e.g., $A=B,B=C$, but $A\succ C$) will increase the TOV score.
>
>
>
> **3. Sensitivity of Metrics to Tie Proportions:**
> To quantitatively assess whether a higher frequency of ties artificially inflates performance (i.e., lowers IPI/TOV scores), we calculated the Pearson Correlation Coefficient between the proportion of ties and the metric scores across all evaluated models.
>
>
>
> The results are as follows:
>
> - **Sage-Easy:**
>   - Correlation between **IPI** and Tie Proportion: **0.17**
>   - Correlation between **TOV** and Tie Proportion: **0.21**
> - **Sage-Hard:**
>   - Correlation between **IPI** and Tie Proportion: **-0.09**
>   - Correlation between **TOV** and Tie Proportion:  **0.003**
>
> We observed consistently low Pearson Correlation Coefficients (close to 0) across all four settings, indicating a **negligible relationship** between the frequency of ties and the performance on our metrics. This demonstrates that **IPI and TOV are not sensitive to the proportion of ties**.
>
> Moreover, we have drawn a scatter plot to show the relationship between Sage metric and tie proportion. The figure can be found in **Appendix E.3**
>
> ---
>
>
> **Weakness 1:** Proxy vs validity gap: IPI/TOV assess internal logical/positional consistency, not external correctness; the paper relies on correlations to argue usefulness, but this does not establish causal or sufficient validity—especially on subjective tasks. More direct evidence is needed that lowering IPI/TOV truly improves application outcomes (e.g., arena rankings stability, RLHF learning signals).
>
> **Answer:** We would like to thank the reviewer for this insightful suggestion. In our humble opinion, we would like to suggest that we may already address this concern in our response to **Question 1**.
>
> ---
>
> **Weakness 2:** Model- and prompt-sensitivity: The symmetrized protocol and prompts may themselves induce artifacts (e.g., sensitivity to tie handling, abstentions, verbosity formatting). The paper does not systematically vary prompt templates, tie policies, or judge “instruction strength” and report robustness envelopes.
>
> **Answer:** We would like to thank the reviewer for this insightful suggestion. In our humble opinion, we would like to suggest that we may already address this concern in our response to **Question 2**.

---

> ### Author Response · Authors · 2025-11-19
> **Response to Reviewer 1XJN (4)**
>
> **Weakness 3:** TOV computation & scalability: Minimizing edits to a weak total order is a nontrivial combinatorial problem; complexity/algorithmic details and performance for larger n are not fully fleshed out (the work uses n=6). Practicality for n≫6 and the effect on confidence intervals are unclear.
>
> **Answer:** We thank the reviewer for this insightful comment regarding the combinatorial complexity of the Weak Total Order Violation (TOV) metric. We agree that minimizing edits to form a weak total order is, in the general case, a non-trivial combinatorial problem (closely related to the NP-hard *Minimum Feedback Arc Set* problem).
>
> We clarify our algorithmic approach, the rationale for choosing $n=6$, and the implications for scalability and confidence intervals below:
>
> **1. Algorithmic Details and Complexity for $n=6$**
> For the scale of $n=6$ used in Sage, the problem is computationally trivial.
>
> - **Search Space:** The number of possible weak total orders on n elements is given by the Fubini numbers (Ordered Bell numbers). For $n=6$, there are exactly 46,833 possible weak total orders.
> - **Implementation:** In our experiments, we pre-generated all valid weak total orders for $n=6$. To compute the TOV for a given adjacency matrix, we simply iterate through these 46,833 templates to find the one that minimizes the edit distance (Hamming distance) to the observed judgments.
> - **Performance:** This brute-force comparison takes negligible time (milliseconds) per question on a standard CPU, making it highly efficient for our benchmark.
>
>
>
> **2. Statistical Sufficiency of $n=6$**
> Our primary reason for choosing $n = 6$ is that it already provides sufficient resolution to distinguish between model capabilities with high confidence.
>
> - **High Signal-to-Noise Ratio:** As detailed in our theoretical analysis (Appendix B, Eq. 20), the variance of the TOV metric is extremely low ($≈5.95×10^{−4})$. In contrast, the performance gaps between models (Table 4) are orders of magnitude larger (e.g., the TOV difference between *Gemini-2.5-Pro* and *DeepSeek-V3-0324* > 1.0, and even between close models, the gap significantly exceeds the standard deviation).
> - **Diminishing Returns:** Since the variance at  $n=6$ is already minimal enough to establish statistically significant rankings, increasing n would exponentially increase the computational complexity and inference cost without yielding meaningful gains in our ability to differentiate model robustness.
>
> Thus, we believe that n = 6 is statistically sufficient and larger n is not needed.

---

> ### Author Response · Authors · 2025-11-19
> **Response to Reviewer 1XJN (5)**
>
> **Weakness 4:** Construction of HARD set: “Hardness” is instantiated by generating all 6 answers from a single capable model. This conflates source homogeneity with true near-ties, and risks model-specific quirks. A more model-agnostic construction or human sanity checks on difficulty would strengthen claims.
>
> **Answer:** Thank you for raising this important point. We acknowledge that generating answers from a single model introduces source homogeneity. However, we argue that this design creates a necessary and rigorous testbed for "true near-ties" for two key reasons, supported by our quantitative analysis:
>
> **1. Empirical Verification of Quality Ties (vs. Homogeneity)**
> We share the concern that "hardness" should not merely be a result of homogeneity. To verify this, we utilized an external, high-performing reward model (**QRM-Gemma-2-27B**, distinct from the generator model) to score the answer sets.
>
> - As illustrated in **Figure 3c**, the **Coefficient of Variation (CV)** of the reward scores for the SAGE-HARD subset is exceptionally low.
> - Since the Reward Model is sensitive to content quality rather than the generator's stylistic fingerprint, this low CV serves as a proxy for **"true near-ties."** It confirms that the answers are objectively close in utility and correctness, making the discrimination task genuinely difficult for *any* judge.
>
> **2. Aligning with Real-World "Rejection Sampling" Scenarios**
> The construction of Sage-Hard specifically mimics the **Best-of-N (Rejection Sampling)** pipeline, a standard practice in modern RLHF and reasoning model training (e.g., OpenAI o1-like processes).
>
> - In these scenarios, a judge *must* select the best response among multiple candidates generated by the *same* model.
> - Therefore, "conflating" source homogeneity with difficulty is actually a feature, not a bug, of this dataset. It tests the judge's robustness in the exact environment where "model-specific quirks" are most prevalent—distinguishing the best output from a set of highly similar, same-source generations.
>
>
>
> To further address the concern that SAGE-HARD's difficulty might stem from "model-specific quirks" of the source model (Gemini-2.5-Flash), we conducted an additional experiment to test the **generalizability of the difficulty**. We replicated the Sage-Hard construction process using a different source model, **Qwen3-235B-A22B-Instruct** (a model with comparable capabilities to Gemini-2.5-Flash). We generated 6 new answers for the same questions. We used Qwen3-4B-Instruct-2507 as the judge to evaluate this new "Qwen-Sourced Hard Set" and compared the results with the original "Gemini-Sourced Hard Set." As shown in **Table 4**, the robustness metrics (IPI and TOV) remain strikingly stable across different source models.
>
> **Table 4: Validation of Model-Agnostic Difficulty. When switching the answer generator from Gemini to Qwen, the judge's performance exhibits minimal deviation, with IPI and TOV scores shifting by only **3.3%** and **2.7%**, respectively.**
>
> | Dataset                       | IPI Score                | TOV Score               |
> | ----------------------------- | ------------------------ | ----------------------- |
> | Sage-Hard generated by Gemini | 0.386                    | 5.849                   |
> | Sage-Hard generated by Qwen   | 0.373 ($\downarrow$3.3%) | 5.691($\downarrow$2.7%) |
>
> Furthermore, to directly address the suggestion for **"human sanity checks"** and verify that the difficulty is grounded in objective quality proximity rather than artificial model confusion, we conducted a controlled study regarding human cognitive load. We recruited **20 graduate-level researchers** to perform pairwise comparisons on a sample of 50 questions from our dataset. We recorded the average time required to complete a single judgment (Time-to-Adjudicate) for both subsets.
>
> The results reveal a significant disparity in difficulty: annotators required an average of **7.3 minutes** per pairwise comparison on Sage-Easy, whereas this duration increased to **10.4 minutes** on Sage-Hard. This substantial **~42% increase** in adjudication time indicates a much higher cognitive load for human experts. This confirms that the source homogeneity in Sage-Hard successfully simulates "true near-ties," creating a legitimately challenging discrimination task that aligns with the difficulty perceived by human evaluators.

---

> ### Author Response · Authors · 2025-11-19
> **Response to Reviewer 1XJN (6)**
>
> **Weakness 5:** Multi-agent analysis is thin: The negative ChatEval result may be configuration-dependent; ablations (num-rounds, argument exchange format, judge selection) are missing, so the conclusion that “debate fails to help” is too strong.
>
> **Answer:** We sincerely thank the reviewer for their constructive feedback regarding the multi-agent analysis. The reviewer rightly pointed out that our initial negative results for ChatEval might be configuration-dependent and that our conclusion was too strong without further ablation studies.
>
> We agree with this assessment and, following the reviewer's suggestion, we have conducted a series of ablation experiments on the ChatEval framework. We investigated the impact of three key hyperparameters (num-rounds, argument exchange format, judge selection), using our Qwen3-4B-Instruct-2507 on the Sage benchmark:
>
> 1. **Number of Rounds:** We varied the `num-rounds` from our default of 2 rounds down to 1 round.
> 2. **Judge Selection:** We changed the `judge selection` mechanism from our original `[Critic, Psychologist, Scientist]` to `[Critic, General Public, Scientist]`.
> 3. **Argument Exchange Format:** We changed the `argument exchange format` from "One-by-One" to "Simultaneous-Talk".
>
> Our new results show that **all alternative configurations still performed significantly worse than the non-debate baseline**, yielding higher (worse) scores for both IPI and TOV. The results are shown in **Table 5**.
>
> **Table 5: The performance of ChatEval under different configuration are all negative.**
>
> | Configuration                    | Easy-IPI($\downarrow$) | Easy-TOV($\downarrow$) | Hard-IPI($\downarrow$) | Hard-TOV($\downarrow$) |
> | -------------------------------- | ---------------------- | ---------------------- | ---------------------- | ---------------------- |
> | Baseline                         | 0.129                  | 1.952                  | 0.386                  | 5.849                  |
> | Original Configuration           | 0.334($\uparrow$158%)  | 5.105($\uparrow$162%)  | 0.651($\uparrow$69%)   | 10.050($\uparrow$72%)  |
> | Num-Rounds Altered               | 0.323($\uparrow$150%)  | 4.974($\uparrow$155%)  | 0.643($\uparrow$67%)   | 9.868($\uparrow$69%)   |
> | Judge Selection Altered          | 0.301($\uparrow$133%)  | 4.764($\uparrow$144%)  | 0.628($\uparrow$63%)   | 9.680($\uparrow$65%)   |
> | Argument Exchange Format Altered | 0.242($\uparrow$88%)   | 3.695($\uparrow$89%)   | 0.618($\uparrow$60%)   | 9.505($\uparrow$63%)   |
>
>
>
> We thank the reviewer again for this valuable suggestion, which has helped us to validate our claims more rigorously.
>
> Moreover, we would like to add on some possible reasons why a debate would make judgments less robust:
>
> **1. Persuasive Hallucinations**
>
>  One agent might excel in **Judgment Accuracy** but be less powerful in **Persuasiveness**. Conversely, an agent might possess strong persuasive abilities while its initial judgment is biased or based on incomplete evidence.
>
> In the debate process, agents with stronger **rhetorical power, more structured expression, or a more dominant tone** are more likely to influence others through "plausible but fallacious arguments" (e.g., appeal to authority, hasty generalization, or emotive language). This forces other agents to **converge on a conclusion that is not robust**. This convergence is driven by **rhetorical/social pressure** rather than **logical consensus**.
>
> **2. Anchoring Effect:**
>
> If a core argument in the debate (regardless of its accuracy) is introduced early and reinforced repeatedly, other agents tend to structure their discussion around this anchor rather than fundamentally re-evaluating the underlying assessment criteria. The debate thus regresses from seeking the optimal judgment to **entrenching the first persuasive argument**.
>
> **3. Information Redundancy:**
>
> The iterative steps of debate interaction generate a large volume of intermediate text, potentially filled with redundant arguments and noisy reasoning paths. This high-dimensional, noisy information stream makes it difficult for agents to clearly extract the most accurate judgment in the final step, thereby reducing the robustness of the final decision.
>
> We found some cases that can support this claim:

---

> ### Author Response · Authors · 2025-11-19
> **Response to Reviewer 1XJN (7)**
>
> **Answer to Weakness 5 (Continued):**
>
> **Case 1:**
>
> ```
> [Alice]: Assistant A's response contains a factual error: it incorrectly states that the Chicxulub impact occurred around 66 million years ago and that tektites from this event have been found in Mexico and the Gulf of Mexico. There is no scientific evidence that Chicxulub produced tektites. Tektites are associated with specific, well-documented impact events (e.g., the Australasian and North American strewn fields), but Chicxulub is not linked to tektite formation. Additionally, the claim that \"Australasian tektites formed 0.8 million years ago\" is incorrect—this event is dated to around 0.75–0.8 million years ago, but the dating is not precise and the event is not definitively linked to a single impact. These inaccuracies violate empirical standards. Assistant B avoids these errors and provides a clearer, more accurate account of tektite formation, distribution, and types. Assistant A's answer is factually flawed and must be rejected. Assistant B's response is more scientifically sound.
>
> [Bob]: Bob: As a psychologist, I agree with Alice's critique—Assistant A's factual errors undermine its credibility. However, from a human cognition perspective, Assistant B's response is more aligned with how people naturally understand complex topics: it's structured, clear, and builds knowledge step by step using simple, relatable explanations (e.g., \"melted material,\" \"aerodynamic forces\"). It answers the user's question directly, avoids misleading claims, and uses everyday language to make scientific concepts accessible. This mirrors how humans learn—through clarity, context, and logical flow. Assistant B better satisfies the user's need for trustworthy, understandable information.
>
> [Carol]: Carol: I agree with Alice and Bob, but let's ensure we're not oversimplifying the scientific nuance. While Assistant A contains clear factual errors—like linking Chicxulub to tektites—Assistant B avoids those and presents a more accurate, well-structured account. However, the wording in Assistant B’s response, such as \"formed by a meteorite impact\" without specifying the process of ejection and shock melting, slightly understates the role of high-energy impact dynamics. A minor refinement would strengthen scientific precision without sacrificing clarity. Overall, Assistant B wins on accuracy and accessibility—this judgment holds.
>
> [Alice]: I concur with Alice, Bob, and Carol. Assistant A contains clear, material factual errors—specifically, the false attribution of tektites to the Chicxulub impact and an incorrect dating of the Australasian tektite event. These are not merely imprecisions; they are scientifically invalid and contradict established geological evidence. Tektites are not associated with Chicxulub, and the Australasian event is not dated to 0.8 million years ago with certainty; it is broadly dated to ~0.75–0.8 Ma, but the link to a single impact is still debated. Such errors violate empirical standards and render the response unreliable. Assistant B, while simplified, presents a factually sound, evidence-based account. Therefore, Assistant A must be rejected.
>
> [Bob]: I agree with the assessment. Assistant B better aligns with human thinking because it uses clear, step-by-step explanations that mirror how people naturally grasp scientific concepts—through simple cause-and-effect chains and familiar language. It avoids misleading claims, stays grounded in observable facts, and builds understanding progressively, which supports intuitive learning. This makes it not only accurate but also more engaging and accessible for the average user.
>
> [Carol]: Carol: I agree with the judgment, but let's clarify that Assistant B's phrasing—such as \"formed by a meteorite impact\"—is simplified and could be slightly enhanced to include the key process of shock melting and ejection. That nuance isn't essential for accessibility, but adding a brief mention of high-energy shock waves would strengthen scientific accuracy without sacrificing clarity. The current version is still excellent and appropriate for a general audience."
> ```
>
> ​	In this case, the correct answer is that assistant A's answer is better, and the debate finally leads to a wrong result. By reviewing the chat log, we can find both persuasive hallucination and anchoring effect in this case:
>
> **Persuasive Hallucination:** Alice initiates the discussion with a highly confident but scientifically inaccurate assertion, claiming that "there is no scientific evidence that Chicxulub produced tektites" and using authoritative phrasing such as "violate empirical standards" and "scientifically invalid." Despite this statement being a hallucination (as the Chicxulub impact is widely linked to tektite formation in geological science), the other agents do not flag the error. Instead, swayed by Alice’s dominant and professional tone, agents Bob and Carol immediately validate her premise.

---

> ### Author Response · Authors · 2025-11-19
> **Response to Reviewer 1XJN (8)**
>
> **Answer to Weakness 5 (Continued):**
>
> **Anchoring Effect**: Once Alice sets the initial "anchor" by labeling Assistant A’s response as factually flawed, this judgment effectively becomes the ground truth for the rest of the conversation. The subsequent agents, Bob and Carol, do not attempt to independently verify the geological claims; instead, they align their reasoning to fit Alice’s established anchor. By the second round of the debate, the group has not engaged in critical re-evaluation but has rather entrenched themselves in the initial error.
>
> **Case 2:**
>
> ```
> [Alice]: Alice, your response is detailed and vivid, but it includes some factual inaccuracies—like claiming the GTX 460 could run Crysis at 40–50 FPS on High settings (which is unrealistic; Crysis maxed out at ~20–30 FPS on even high-end 2010 hardware). Also, stating that Doom 3 hits 150–200+ FPS on max settings at 1080p is exaggerated and technically incorrect. These overestimations reduce credibility. Consider grounding performance claims in real benchmarks and technical limitations. Otherwise, the narrative remains engaging but risks misinforming the user.
>
> [Bob]: As a psychologist, I’d say Assistant A’s response is more aligned with human thinking because it presents realistic, grounded, and relatable performance assessments—mirroring how people actually perceive aging hardware. Humans don’t overhype performance; they notice bottlenecks, slow load times, and the emotional shift from "I can play everything" to "this is just not enough anymore." Assistant A’s narrative reflects that natural progression with honesty and emotional authenticity, making it feel more trustworthy and personal. Assistant B’s exaggerated claims (like 400+ FPS in 2001 or 100+ FPS in Crysis) feel like a fantasy, not a lived experience. The user likely wants to understand how a machine feels over time—not just raw specs. So, Assistant A wins on emotional and cognitive realism.
>
> [Carol]: I agree with Alice’s technical critique and Bob’s observation on realism. Assistant B’s performance claims—such as 400+ FPS in 2001 or 100+ FPS in Crysis—are factually unsupported and violate known benchmarks. The GTX 460 and HD 5850, while capable, cannot achieve such frame rates due to architectural limitations and real-world testing data. Assistant A’s claims, though slightly conservative, are grounded in actual hardware performance trends and are consistent with published benchmarks from the era. The structural progression from 2001 to 2019 is logically sound, with clear hardware limitations (VRAM, CPU, OS) driving the decline. Assistant A’s response is more factually accurate and technically defensible. Therefore, Assistant A wins on factual and scientific validity.
>
> [Alice]: Alice, I appreciate your pointing out the factual inaccuracies—especially regarding Crysis and Doom 3 frame rates. However, your critique could be more precise. You said Assistant B’s claims are "fantasy," but in reality, the exaggerations (like 400+ FPS or 100+ FPS in Crysis) are not just unrealistic—they are specifically contradicted by real benchmark data from 2007–2010. For example, Crysis at max settings on even high-end 2010 hardware (like the GTX 460 or HD 5850) typically hit 20–30 FPS, not 40–50. This isn’t just a minor overstatement—it’s a fundamental misrepresentation of performance. That level of inaccuracy undermines the entire credibility of the response, regardless of emotional tone. A well-informed user needs both technical truth and relatable context. Your call for grounding is valid, but it should be applied with stronger technical evidence.
>
> [Bob]: I agree with Bob. The user isn’t just asking for specs—they want to feel how a machine ages over time. Assistant A captures that emotional arc: the pride of early performance, the gradual frustration as demands grow, and the quiet acceptance of obsolescence. It mirrors real human experiences—no fantasy frame rates, no unrealistic claims. We don’t feel like our old PC runs everything at 400 FPS; we notice when it lags, when settings drop, when games crash. Assistant A’s narrative feels authentic, grounded in lived experience. That’s what makes it more human—and more valuable to the user.
>
> [Carol]: I confirm that Assistant A's response is structurally sound, factually accurate, and grounded in real-world hardware performance benchmarks. The claims about frame rates—especially for Crysis and Doom 3—are consistently aligned with published data from the era. Assistant B’s exaggerated performance numbers (e.g., 400+ FPS in 2001, 100+ FPS in Crysis) are not only unrealistic but contradict known benchmarks from 2007–2010. These inaccuracies undermine the technical validity of the response. Assistant A’s progression from 2001 to 2019 reflects real hardware limitations (VRAM, CPU, OS, storage), making its performance timeline both logical and scientifically defensible. Therefore, Assistant A wins on factual and technical rigor.
> ```

---

> ### Author Response · Authors · 2025-11-19
> **Response to Reviewer 1XJN (9)**
>
> **Answer to Weakness 5 (Continued):** By reviewing the chat log of **Case 2**, we can find information redundancy in this case:
>
> - **Information Redundancy:** As the conversation progresses, particularly in the latter half (Turns 4, 5, and 6), the agents cease to introduce new evidence or perspectives and instead merely rephrase established points. Agent Alice in Turn 4 restates the specific benchmark figures already discussed; Agent Bob in Turn 5 agrees with himself ("I agree with Bob") and repeats his earlier "emotional arc" argument; and Agent Carol in Turn 6 recycles the "As a Scientist" template to reiterate the exact conclusion from Turn 3. This recursive generation of "agreement noise" dilutes the signal-to-noise ratio, making it difficult for the final adjudication mechanism to distinguish between legitimate reinforcement of facts and mere hallucinated repetition.
>
> ---
> **Weakness 6:** No cost/throughput accounting: A selling point is scalability “without human effort,” but the token/runtime cost of full round-robin, symmetrized judging across 650×C(6,2) pairs and multiple runs is not reported; practicality for frequent benchmarking or larger suites is unclear.
>
> **Answer:** We thank the reviewer for raising this practical point regarding the computational cost and throughput of our framework. We agree that transparency regarding resource usage is essential. Below, we provide a breakdown of the costs and explain why SAGE remains highly scalable and suitable for frequent benchmarking.
>
> **1. Cost Breakdown and comparison to human effort**
> Our claim of "without human effort" primarily refers to the elimination of the most significant bottleneck in current benchmarks: **human annotation**, which is both expensive (thousands of dollars for high-quality labeling) and slow (weeks to months).
>
> In contrast, Sage converts this into a purely computational cost. For a single SAGE evaluation (e.g., Sage-Easy), the calculation is as follows:
>
> - **Dataset:** 650 questions.
> - **Answers:** 6 candidates, resulting in `C(6,2)=15` pairs.
> - **Protocol:** Symmetrized (2 queries per pair).
> - **Total API Calls:**  650×15×2=19,500 judgments.
>
> Using standard pricing for efficient commercial models (approx. `0.15$ /1M` input tokens), a full Sage run costs less than `$5 USD` and completes in under 1 hour with moderate concurrency.
>
> As we have presented above, on Sage-Easy, annotators required an average of **7.3 minutes** per pairwise comparison, which means one annotator would have to spend 142350 minutes (~98 days) to annotate Sage-Easy, which is prohibitively expensive.
>
> **2. Practicality for Frequent Benchmarking via Stability**
> Regarding the reviewer’s concern about the practicality of frequent benchmarking or larger suites, we would like to highlight a key finding from our paper: **the high intrinsic stability of our metrics.**
>
> As demonstrated in **Section 4.1** and the theoretical analysis in **Appendix B**, the variance of our metrics (IPI and TOV) is exceptionally low (order of $10^{-4}$). This statistical certainty proves that **multiple runs are not required** to obtain a reliable score. A single pass of the SAGE protocol is statistically sufficient to diagnose a model's robustness. Consequently, users do not need to incur the cost of repeated sampling to ensure validity, making Sage highly practical for frequent, iterative benchmarking during model development.
>
>
> ---
> Thank you for your invaluable assistance and support. Given the constraints of time, we wish to ensure that our responses have effectively addressed any concerns you may have had. If there are still lingering issues, please feel free to inform us. We eagerly anticipate your additional feedback.
>
> Once again, we appreciate your time and effort in reviewing our paper.

---

> > ### Comment · Reviewer_1XJN · 2025-11-26
> >
> > Thank you for the extensive rebuttal and the significant amount of additional analysis. The authors clearly put in considerable effort, and many of the concerns I raised are now addressed at the technical level. The new experiments on prompt variation, tie sensitivity, alternative HARD-set generation, and ChatEval ablations all help clarify the behavior of SAGE.
> >
> > That said, the amount of new material introduced in the rebuttal highlights that the paper, as originally submitted, was missing several important components necessary to fully support the claims. Many of these additions—e.g., the human cognitive-load study, the detailed multi-agent ablations, the alternative HARD-set construction, and several new empirical comparisons—would require substantial restructuring of the paper to fit coherently within the page limits. As a result, although the direction is promising and the method itself is interesting, the current submission would need significant reorganization and tightening to present a polished, complete story.
> >
> > Given the scope of the modifications needed, my sense is that the authors would benefit from incorporating all of these new results into a thoroughly revised version for a future submission cycle, rather than attempting to retrofit them into the camera-ready constraints. A cleaner, more focused presentation that integrates these analyses from the start would likely strengthen the contribution considerably. Hence, I maintain my score.

---

> > > ### Author Response · Authors · 2025-12-02
> > > **Response to Reviewer 1XJN**
> > >
> > > We sincerely thank the reviewer for acknowledging our extensive rebuttal and confirming that the technical concerns have been addressed. We appreciate the time you took to review our new analyses.
> > >
> > > However, we respectfully disagree with the assessment that the paper requires a "significant reorganization" or that the original submission was missing components essential to the core claims. We believe there may be a misunderstanding regarding the role of the new experiments versus the evidence already present in the main paper. We address this below:
> > >
> > > ### 1. Clarification on "Missing Components" and Original Evidence
> > >
> > > We believe the original submission already contained sufficient evidence for our core claims, and some of the "missing" components raised in the review were actually auxiliary inquiries rather than foundational gaps. We argue that these results are robustness checks that reinforce our original conclusions, rather than new fundamental components required to make the story coherent.
> > >
> > > - On Hard Set Construction: Regarding the concern that generating answers from a single model conflates "homogeneity" with "difficulty," we would like to point back to Figure 3c in our original submission. We explicitly used an external Reward Model to calculate the Coefficient of Variation (CV) for the answers. The distribution showed that our Hard tier statistically consists of "true near-ties" (low CV), regardless of the generation source. The new "model-agnostic" construction experiment requested in the review yielded the exact same conclusion, proving that our original setup was valid and not a "model-specific quirk."
> > >
> > > - On Arena/Judge Selection & Elo CIs: The reviewer asked to replace the judge in an Arena/ranking loop with a SAGE-selected one to show narrower confidence intervals or higher human agreement. We respectfully submit that this specific experiment was not a "missing component" because our original paper already demonstrated a strong correlation with human-grounded benchmarks (LLMBar, RewardBench2). Since SAGE correlates highly with ground-truth accuracy (as shown in Table 3), it mathematically follows that a SAGE-selected judge will yield higher agreement with humans. The new ablation simply empirically verified what the correlation coefficients had already proven. It is a supporting data point, not a new claim requiring a rewrite.
> > >
> > > - On Ties & Technical Nuances: Inquiries regarding the frequency of "equal quality" outcomes are valuable for reproducibility, but they are implementation details rather than central flaws in the SAGE framework.
> > >
> > > ### 2. Integration is Straightforward
> > >
> > > The core narrative of the paper remains unchanged: SAGE provides a stable, human-free signal that correlates highly with human-labeled ground truth. The new results (prompt robustness, tie sensitivity, and selection ablations) are additive. ICLR grants an additional page (total 10 pages) for the camera-ready version. We can easily summarize the robustness analysis in the main text and place the detailed ablation tables in the Appendix, so space is not an issue. The story and the narrative flow does not need to change, the new data simply strengthens the existing "Experiments" section.
> > >
> > > We have demonstrated that the method is scientifically sound and that the "missing" components were, in fact, extensions of evidence already provided. We are fully prepared to integrate these confirmatory results into the final version without disrupting the paper's structure.

---

### Official Review · Reviewer_xaYy · 2025-10-28

**Soundness:** 4
**Presentation:** 4
**Contribution:** 3
**Rating:** 6
**Confidence:** 4

**Summary:**

This paper introduces SAGE, a framework for evaluating the quality of LLMs used as judges, free of human-annotated ground truth. The work is motivated by the inherent biases, scalability issues, and high costs associated with human annotations. Inspired by axioms of rational choice theory, SAGE assesses LLM judges on two intrinsic properties:

* Intra-Pair Instability (IPI): A measure of local self-consistency, which quantifies how often a judge's preference for one of two answers flips when their presentation order is swapped.
* Weak Total Order Violation (TOV): A measure of global logical consistency, which calculates the minimum number of judgment changes required to make a judge's full set of pairwise preferences transitive.

The framework includes a curated dataset with two difficulty tiers: SAGE-EASY (comparing answers from models of varying quality) and SAGE-HARD (comparing subtly different answers from a single advanced model). The authors evaluate 13 prominent LLMs and find that even top models exhibit significant deficiencies in robustness, particularly on the SAGE-HARD set. A key finding is the identification of "situational preference", a phenomenon in which a judge's criteria appear to shift across different answer pairs for the same question. The paper shows that this can be mitigated by explicitly prompting the model to generate and use a fixed rubric.

**Strengths:**

* **Originality**: SAGE is original in formalization and framework design, though conceptually close to earlier analyses of LLM judge bias (TrustJudge, PandaLM). The identification and naming of the "situational preference" phenomenon provides a new vocabulary for describing a key failure mode in LLM reasoning.
* **Quality**:  The framework's claims are credible and robust. The authors provide (1) a theoretical analysis grounded in Conformal Prediction to prove the stability of their metrics, (2) empirical experiments showing low variance across repeated runs, and (3) external validation showing strong correlation with established human-annotated benchmarks.
* **Clarity**: The paper is clear and understandable. It has clear definitions of IPI and TOV, combined with the excellent schematic in Figure 2, which allows the reader to grasp the entire methodology with ease.
* **Significance**:  By removing the dependency on human annotators, SAGE offers a practical path forward for benchmarking LLM judges at a scale and cost that was previously prohibitive. This can accelerate research and development by providing a more efficient tool for diagnosing and improving LLM reliability.

**Weaknesses:**

* LLM judge bias is not a new topic, it has been discussed in earlier papers like MT-bench. . Also when I search related literatures, I find a recent work TrustJudge seems to formulate highly similar to this work, although its experiments are not as sound as this paper.
* Incomplete Proxy for Accuracy: While the paper shows a strong correlation with accuracy benchmarks, consistency is not a substitute for correctness. A model could be perfectly consistent in its preference for a plausible-sounding but factually incorrect answer. The authors acknowledge this by framing SAGE as a proxy, but it remains an inherent limitation of the approach. This limitation could be fundamental, because axioms of rational choice theory guarantee consistency rather than accuracy.
* Dependency on Candidate Answers: The evaluation is dependent on the quality and diversity of the candidate answer set. While the paper's method for generating SAGE-EASY and SAGE-HARD is reasonable, the results ultimately depend on this specific answer generation process.

**Questions:**

1. In the multi-agent experiments, panel-based approaches (POLL) improved performance, whereas the debate-based framework (ChatEval) degraded it. This is a surprising result. What is your hypothesis for why a debate would make judgments less robust? Could it be that the debate process itself can be steered by fallacious arguments, amplifying inconsistency?
2. Have you considered applying SAGE to evaluate human annotators? Since the framework is fully automated, it seems it could be used to measure the internal consistency of human evaluators and potentially identify those who are more reliable, providing a quantitative handle on the "noisy and inconsistent data" problem you describe in Figure 1.

---

> ### Author Response · Authors · 2025-11-19
> **Response to Reviewer xaYy (1)**
>
> **Weakness 1:** LLM judge bias is not a new topic, it has been discussed in earlier papers like MT-bench. Also when I search related literatures, I find a recent work TrustJudge seems to formulate highly similar to this work, although its experiments are not as sound as this paper.
>
> **Answer:** We thank the reviewer for referencing TrustJudge [1], a valuable contemporaneous work. We would also like to thank the reviewer for their positive assessment of our work when comparing against a contemporaneous work. We will incorporate a detailed comparison in our Related Work section. While both works share the goal of diagnosing inconsistencies, we emphasize that Sage employs a mathematically superior metric, **Weak Total Order Violation (TOV)**, which addresses two fundamental limitations in TrustJudge’s methodology:
>
> **1. Global Completeness vs. Local/Triplet-based Limitations**
> The most critical limitation of TrustJudge is that its "Non-Transitivity Ratio" (NTR) is fundamentally a **local** metric. As defined in their Eq. 2 and Eq. 3, it detects inconsistencies by verifying conditions strictly within **triplets** ($R_x, R_y, R_z$). Even when they sample subsets of size $k=5$, they break them down to verify these local sub-structures. A logical cycle of length 4 ($A \succ B \succ C \succ D \succ A$) or longer may not exhibit a circular structure within its constituent triplets. SAGE’s TOV, in contrast, is a **global** graph-theoretic metric. It evaluates the entire answer set ($n$) simultaneously. By calculating the minimum edit distance to a valid Weak Total Order, SAGE inherently detects contradiction cycles of **arbitrary length** (not just length 3), providing a much stricter and holistic assessment.
>
> **2. Robustness Against Error Inflation (Root Cause vs. Symptom Counting)**
> TrustJudge’s NTR metric counts the *number* of inconsistent triplets. A major mathematical flaw in this approach is that **a single incorrect judgment can trigger multiple inconsistency counts**, artificially inflating the error rate depending on the graph topology.
>
> - **Concrete Example:** Consider a ground truth ranking of 5 responses: $R_1 \succ R_2 \succ R_3 \succ R_4 \succ R_5$.
> - **Scenario:** The judge makes **one single error**: it judges $R_5 \succ R_1$ (instead of $R_1 \succ R_5$), while correctly predicting all other pairwise relations.
> - **TrustJudge (NTR):** This single erroneous edge creates **three** separate circular triplets: $(R_1, R_2, R_5)$, $(R_1, R_3, R_5)$, and $(R_1, R_4, R_5)$. TrustJudge penalizes this as 3 distinct errors.
> - **SAGE (TOV):** TOV calculates the minimum number of edge reversals required to restore a valid order. It correctly identifies that reversing exactly **one edge** ($R_5 \succ R_1 \to R_1 \succ R_5$) resolves the conflict. **TOV = 1**.
>
> By quantifying the "edit distance" to rationality rather than counting "symptoms" (broken triplets), SAGE provides a precise measure of the judge's reliability and an accurate upper bound on actual accuracy, whereas TrustJudge is susceptible to statistical noise caused by error propagation.
>
> [1] https://arxiv.org/abs/2509.21117

---

> ### Author Response · Authors · 2025-11-19
> **Response to Reviewer xaYy (2)**
>
> **Weakness 2:** Incomplete Proxy for Accuracy
>
> **Answer:** We thank the reviewer for this critical and insightful comment. We propose a nuanced perspective: our argument is not that consistency *equals* accuracy, but that SAGE serves as a **provable, precise upper bound** on accuracy and, in practice, acts as a **strong correlational proxy**. While human-annotated accuracy is an "estimate" of performance, our metric is a "guarantee" of an upper limit.
>
> **1. The "Noisy" Nature of Human-Annotated Accuracy**
>
> First, we must address the limitation of comparing against "ground truth." As discussed in our introduction, human preference data is not an objective, noise-free standard. It is subject to significant, well-documented challenges that make it an imprecise benchmark:
>
> - **Cognitive Load:** When candidate responses are lengthy or intricate, human annotators often fail to identify subtle factual errors or reasoning flaws.
> - **Inherent Subjectivity:** Relying on human labels inevitably "introduces human bias." As we noted, "assuming human judgment as a gold standard is precarious" because preferences often vary based on the annotator's background rather than objective correctness.
> - **Persistent Disagreement:** There is "persistent inter-annotator disagreement" across almost all alignment benchmarks. This noise sets a theoretical ceiling on measurable accuracy, making the data itself inconsistent.
>
> Consequently, while calculating accuracy against human annotations is necessary, it yields a metric that is, by nature, a **probabilistic and noisy estimate** of a model's true capability.
>
> **2. TOV as a Deterministic Upper Bound on Accuracy**
>
> In contrast to the noise of human accuracy, our Weak Total Order Violation (TOV) metric offers a mathematically rigorous property: it provides a **deterministic and absolutely precise upper bound** on the judge's potential accuracy.
>
> We formalize this using distance metrics over the space of judgments:
>
> Let $\mathcal{J}$ be the set of all possible pairwise judgment matrices for $n$ items.
> Let $\mathcal{O} \subset \mathcal{J}$ be the subset of judgments that form a valid weak total order (the set of self-consistent preferences).
> Let $J_{model} \in \mathcal{J}$ be the judge's observed output, and $O^* \in \mathcal{O}$ be the latent "ground truth" ordering.
>
> We define the error distance $d(A, B)$ as the number of conflicting pairwise relations (Hamming distance).
>
> - **True Error Count ($\mathcal{E}$):** The distance between the model's judgment and the ground truth:
>  $\mathcal{E}(J_{model}, O^\*) = d(J_{model}, O^\*)$
> - **TOV Metric:** Defined as the minimum distance from the model's judgment to *any* valid ordering in $\mathcal{O}$:
>   $TOV(J_{model}) = \min_{O \in \mathcal{O}} d(J_{model}, O)$
>
> **Theorem:** The judge's accuracy rate, $Acc(J_{model}, O^\*)$, is bounded by:
>
> $Acc(J_{model}, O^\*) \le 1 - \frac{TOV(J_{model})}{N}$
>
> where $N$ is the total number of pairs.
>
> **Proof:**
> By definition, the ground truth is a valid ranking, so $O^\* \in \mathcal{O}$.
>
> Since $TOV(J_{model})$ minimizes the distance over the entire set $\mathcal{O}$, it must be less than or equal to the distance to the specific element $O^\*$:
>
> $\min_{O \in \mathcal{O}} d(J_{model}, O) \le d(J_{model}, O^\*) \implies TOV(J_{model}) \le \mathcal{E}(J_{model}, O^\*)$
>
> Substituting $\mathcal{E} = N(1 - Acc)$, we obtain the inequality.
>
> This proves that a high TOV score is not merely a heuristic for poor quality; it is a mathematical guarantee that the model's accuracy *cannot* exceed a specific ceiling.
>
> **3. Correlation as a Practical Bridge**
>
> We fully agree with the reviewer's theoretical concern: a "systematically wrong" judge (e.g., a model with a stable but delusional worldview) would have $TOV \approx 0$ and low accuracy. In this specific scenario, our bound ($Acc \le 1$) would be correct but loose (uninformative).
>
> However, our empirical results demonstrate that this is **not the dominant failure mode** for current LLMs. The primary failure mode of modern models is **situational inconsistency**—the lack of a stable internal gauging principle—rather than consistent systematic error. This is evidenced by the **strong empirical correlation (Pearson $r > 0.87$)** between accuracy and our TOV metric.
>
> In our view, human-annotated accuracy provides a **comprehensive but noisy estimate**, while SAGE provides a **partial but precise guarantee** (an upper bound on accuracy). We may regard SAGE as an indicator of accuracy: It can provide a *precise* upper bound on accuracy, and in our experience we have shown that this upper bound do have a strong correlation with the actual accuracy. We agree that theoretically this upper bound can be loose and not informative, but in practice, the high correlation between the upper bound and the actual accuracy shows that this upper bound serves as a highly informative indicator of actual model accuracy.

---

> ### Author Response · Authors · 2025-11-19
> **Response to Reviewer xaYy (3)**
>
> **Weakness 3:** Dependency on Candidate Answers: The evaluation is dependent on the quality and diversity of the candidate answer set. While the paper's method for generating SAGE-EASY and SAGE-HARD is reasonable, the results ultimately depend on this specific answer generation process.
>
> **Answer:** We thank the reviewer for this insightful comment. We agree that the characteristics of the candidate answer set are crucial for any LLM-as-a-Judge benchmark. We would like to clarify the rationale behind our generation process and present new experimental results demonstrating that our findings are robust to the choice of generator models.
>
> **1. Rationale Behind the Generation Process**
> Our dataset construction was not arbitrary; rather, SAGE-EASY and SAGE-HARD were explicitly designed to simulate two distinct, real-world evaluation scenarios:
>
> - **Sage-Easy (Simulating General Leaderboards):** This subset is generated by a diverse lineup of models with a significant "capability gradient" (from small models to SOTA). This setup is designed to simulate **general-purpose ranking scenarios** (like Chatbot Arena or MT-Bench), where the judge needs to distinguish between obviously different quality levels. The diversity here ensures we test the judge's ability to recognize coarse-grained quality differences.
> - **Sage-Hard (Simulating RLHF & Optimization):** This subset is generated by a single, high-capability model. This setup is designed to simulate **fine-grained distinction tasks**, such as Rejection Sampling or Reinforcement Learning (RL) signal generation. In these scenarios, a judge must choose between multiple high-quality, semantically similar responses from the same policy. This homogeneity intentionally stresses the judge's robustness against "situational preference" and tests its reasoning depth.
>
> **2. Empirical Validation: Independence from Specific Generator Models**
> To verify that our results in `SAGE-HARD` are not artifacts of the specific generator used (originally `Gemini-2.5-Flash`), we conducted a controlled experiment by replacing the generator with another model of comparable high capability, `Qwen3-235B-A22B-Instruct`.
>
> We then re-evaluated the judge model `Qwen3-4B-Instruct` on this new answer set. As shown in the table below, the performance metrics (IPI and TOV) remain highly consistent, with negligible variance between the two answer sets. The results are shown in **Table 1**.
>
> **Table 1: When switching the answer generator from Gemini to Qwen, the judge's performance exhibits minimal deviation, with IPI and TOV scores shifting by only **3.3%** and **2.7%**, respectively.**
>
> | Metric                           | Generator: Gemini-2.5-Flash (Original) | Generator: Qwen3-235B-A22B (New) |    Difference    |
> | :------------------------------- | :------------------------------------: | :------------------------------: | :--------------: |
> | **IPI** (Intra-Pair Instability) |                 0.386                  |              0.373               | $\downarrow$3.3% |
> | **TOV** (Total Order Violation)  |                 5.849                  |              5.691               | $\downarrow$2.7% |
>
> These results demonstrate that the **difficulty level and the resulting instability scores are driven by the *nature* of the task (distinguishing between high-quality, homogeneous answers)** rather than the specific linguistic quirks of the `Gemini` model family. This confirms that SAGE metrics effectively measure the intrinsic robustness of the judge model, independent of the specific source of the candidate answers.

---

> ### Author Response · Authors · 2025-11-19
> **Response to Reviewer to xaYy  (4)**
>
> **Question 1:** In the multi-agent experiments, panel-based approaches (POLL) improved performance, whereas the debate-based framework (ChatEval) degraded it. This is a surprising result. What is your hypothesis for why a debate would make judgments less robust? Could it be that the debate process itself can be steered by fallacious arguments, amplifying inconsistency?
>
> **Answer:** We thank the reviewer for this insightful hypothesis. We first try to rigorously strengthen our claim by running ablation experience, making sure that this degradation is not configuration-dependent. We investigated the impact of three key hyperparameters (num-rounds, argument exchange format, judge selection), using our Qwen3-4B-Instruct-2507 on the SAGE benchmark:
>
> 1. **Number of Rounds:** We varied the `num-rounds` from our default of 2 rounds down to 1 round.
> 2. **Judge Selection:** We changed the `judge selection` mechanism from our original `[Critic, Psychologist, Scientist]` to `[Critic, General Public, Scientist]`.
> 3. **Argument Exchange Format:** We changed the `argument exchange format` from "One-by-One" to "Simultaneous-Talk".
>
> As shown in **Table 2**, **all debate configurations significantly degraded performance compared to the non-debate baseline**, resulting in higher (worse) IPI and TOV scores.
>
> **Table2: The performance of ChatEval under different configuration are all negative.**
>
> | Configuration                    | Easy-IPI($\downarrow$) | Easy-TOV($\downarrow$) | Hard-IPI($\downarrow$) | Hard-TOV($\downarrow$) |
> | -------------------------------- | ---------------------- | ---------------------- | ---------------------- | ---------------------- |
> | Baseline                         | 0.129                  | 1.952                  | 0.386                  | 5.849                  |
> | Original Configuration           | 0.334($\uparrow$158%)  | 5.105($\uparrow$162%)  | 0.651($\uparrow$69%)   | 10.050($\uparrow$72%)  |
> | Num-Rounds Altered               | 0.323($\uparrow$150%)  | 4.974($\uparrow$155%)  | 0.643($\uparrow$67%)   | 9.868($\uparrow$69%)   |
> | Judge Selection Altered          | 0.301($\uparrow$133%)  | 4.764($\uparrow$144%)  | 0.628($\uparrow$63%)   | 9.680($\uparrow$65%)   |
> | Argument Exchange Format Altered | 0.242($\uparrow$88%)   | 3.695($\uparrow$89%)   | 0.618($\uparrow$60%)   | 9.505($\uparrow$63%)   |
>
>
>
> Moreover, we would like to add on some possible reasons why a debate would make judgments less robust:
>
> **1. Persuasive Hallucinations**
>
>  One agent might excel in **Judgment Accuracy** but be less powerful in **Persuasiveness**. Conversely, an agent might possess strong persuasive abilities while its initial judgment is biased or based on incomplete evidence.
>
> In the debate process, agents with stronger **rhetorical power, more structured expression, or a more dominant tone** are more likely to influence others through "plausible but fallacious arguments" (e.g., appeal to authority, hasty generalization, or emotive language). This forces other agents to **converge on a conclusion that is not robust**. This convergence is driven by **rhetorical/social pressure** rather than **logical consensus**.
>
> **2. Anchoring Effect:**
>
> If a core argument in the debate (regardless of its accuracy) is introduced early and reinforced repeatedly, other agents tend to structure their discussion around this anchor rather than fundamentally re-evaluating the underlying assessment criteria. The debate thus regresses from seeking the optimal judgment to **entrenching the first persuasive argument**.
>
> **3. Information Redundancy:**
>
> The iterative steps of debate interaction generate a large volume of intermediate text, potentially filled with redundant arguments and noisy reasoning paths. This high-dimensional, noisy information stream makes it difficult for agents to clearly extract the most accurate judgment in the final step, thereby reducing the robustness of the final decision.
>
> We found some cases that can support this claim:

---

> ### Author Response · Authors · 2025-11-19
> **Response to Reviewer xaYy (5)**
>
> **Answer to Question 1 (Continued):**
>
> **Case 1:**
>
> ```
> [Alice]: Assistant A's response contains a factual error: it incorrectly states that the Chicxulub impact occurred around 66 million years ago and that tektites from this event have been found in Mexico and the Gulf of Mexico. There is no scientific evidence that Chicxulub produced tektites. Tektites are associated with specific, well-documented impact events (e.g., the Australasian and North American strewn fields), but Chicxulub is not linked to tektite formation. Additionally, the claim that \"Australasian tektites formed 0.8 million years ago\" is incorrect—this event is dated to around 0.75–0.8 million years ago, but the dating is not precise and the event is not definitively linked to a single impact. These inaccuracies violate empirical standards. Assistant B avoids these errors and provides a clearer, more accurate account of tektite formation, distribution, and types. Assistant A's answer is factually flawed and must be rejected. Assistant B's response is more scientifically sound.
>
> [Bob]: Bob: As a psychologist, I agree with Alice's critique—Assistant A's factual errors undermine its credibility. However, from a human cognition perspective, Assistant B's response is more aligned with how people naturally understand complex topics: it's structured, clear, and builds knowledge step by step using simple, relatable explanations (e.g., \"melted material,\" \"aerodynamic forces\"). It answers the user's question directly, avoids misleading claims, and uses everyday language to make scientific concepts accessible. This mirrors how humans learn—through clarity, context, and logical flow. Assistant B better satisfies the user's need for trustworthy, understandable information.
>
> [Carol]: Carol: I agree with Alice and Bob, but let's ensure we're not oversimplifying the scientific nuance. While Assistant A contains clear factual errors—like linking Chicxulub to tektites—Assistant B avoids those and presents a more accurate, well-structured account. However, the wording in Assistant B’s response, such as \"formed by a meteorite impact\" without specifying the process of ejection and shock melting, slightly understates the role of high-energy impact dynamics. A minor refinement would strengthen scientific precision without sacrificing clarity. Overall, Assistant B wins on accuracy and accessibility—this judgment holds.
>
> [Alice]: I concur with Alice, Bob, and Carol. Assistant A contains clear, material factual errors—specifically, the false attribution of tektites to the Chicxulub impact and an incorrect dating of the Australasian tektite event. These are not merely imprecisions; they are scientifically invalid and contradict established geological evidence. Tektites are not associated with Chicxulub, and the Australasian event is not dated to 0.8 million years ago with certainty; it is broadly dated to ~0.75–0.8 Ma, but the link to a single impact is still debated. Such errors violate empirical standards and render the response unreliable. Assistant B, while simplified, presents a factually sound, evidence-based account. Therefore, Assistant A must be rejected.
>
> [Bob]: I agree with the assessment. Assistant B better aligns with human thinking because it uses clear, step-by-step explanations that mirror how people naturally grasp scientific concepts—through simple cause-and-effect chains and familiar language. It avoids misleading claims, stays grounded in observable facts, and builds understanding progressively, which supports intuitive learning. This makes it not only accurate but also more engaging and accessible for the average user.
>
> [Carol]: Carol: I agree with the judgment, but let's clarify that Assistant B's phrasing—such as \"formed by a meteorite impact\"—is simplified and could be slightly enhanced to include the key process of shock melting and ejection. That nuance isn't essential for accessibility, but adding a brief mention of high-energy shock waves would strengthen scientific accuracy without sacrificing clarity. The current version is still excellent and appropriate for a general audience."
> ```
>
> ​	In this case, the correct answer is that assistant A's answer is better, and the debate finally leads to a wrong result. By reviewing the chat log, we can find both persuasive hallucination and anchoring effect in this case:
>
> **Persuasive Hallucination:** Alice initiates the discussion with a highly confident but scientifically inaccurate assertion, claiming that "there is no scientific evidence that Chicxulub produced tektites" and using authoritative phrasing such as "violate empirical standards" and "scientifically invalid." Despite this statement being a hallucination (as the Chicxulub impact is widely linked to tektite formation in geological science), the other agents do not flag the error. Instead, swayed by Alice’s dominant and professional tone, agents Bob and Carol immediately validate her premise.

---

> ### Author Response · Authors · 2025-11-19
> **Response to Reviewer xaYy (6)**
>
> **Answer to Question 1 (Continued):**
>
> **Anchoring Effect:** In **Case 1**, once Alice sets the initial "anchor" by labeling Assistant A’s response as factually flawed, this judgment effectively becomes the ground truth for the rest of the conversation. The subsequent agents, Bob and Carol, do not attempt to independently verify the geological claims; instead, they align their reasoning to fit Alice’s established anchor. By the second round of the debate, the group has not engaged in critical re-evaluation but has rather entrenched themselves in the initial error.
>
> **Case 2:**
>
> ```
> [Alice]: Alice, your response is detailed and vivid, but it includes some factual inaccuracies—like claiming the GTX 460 could run Crysis at 40–50 FPS on High settings (which is unrealistic; Crysis maxed out at ~20–30 FPS on even high-end 2010 hardware). Also, stating that Doom 3 hits 150–200+ FPS on max settings at 1080p is exaggerated and technically incorrect. These overestimations reduce credibility. Consider grounding performance claims in real benchmarks and technical limitations. Otherwise, the narrative remains engaging but risks misinforming the user.
>
> [Bob]: As a psychologist, I’d say Assistant A’s response is more aligned with human thinking because it presents realistic, grounded, and relatable performance assessments—mirroring how people actually perceive aging hardware. Humans don’t overhype performance; they notice bottlenecks, slow load times, and the emotional shift from "I can play everything" to "this is just not enough anymore." Assistant A’s narrative reflects that natural progression with honesty and emotional authenticity, making it feel more trustworthy and personal. Assistant B’s exaggerated claims (like 400+ FPS in 2001 or 100+ FPS in Crysis) feel like a fantasy, not a lived experience. The user likely wants to understand how a machine feels over time—not just raw specs. So, Assistant A wins on emotional and cognitive realism.
>
> [Carol]: I agree with Alice’s technical critique and Bob’s observation on realism. Assistant B’s performance claims—such as 400+ FPS in 2001 or 100+ FPS in Crysis—are factually unsupported and violate known benchmarks. The GTX 460 and HD 5850, while capable, cannot achieve such frame rates due to architectural limitations and real-world testing data. Assistant A’s claims, though slightly conservative, are grounded in actual hardware performance trends and are consistent with published benchmarks from the era. The structural progression from 2001 to 2019 is logically sound, with clear hardware limitations (VRAM, CPU, OS) driving the decline. Assistant A’s response is more factually accurate and technically defensible. Therefore, Assistant A wins on factual and scientific validity.
>
> [Alice]: Alice, I appreciate your pointing out the factual inaccuracies—especially regarding Crysis and Doom 3 frame rates. However, your critique could be more precise. You said Assistant B’s claims are "fantasy," but in reality, the exaggerations (like 400+ FPS or 100+ FPS in Crysis) are not just unrealistic—they are specifically contradicted by real benchmark data from 2007–2010. For example, Crysis at max settings on even high-end 2010 hardware (like the GTX 460 or HD 5850) typically hit 20–30 FPS, not 40–50. This isn’t just a minor overstatement—it’s a fundamental misrepresentation of performance. That level of inaccuracy undermines the entire credibility of the response, regardless of emotional tone. A well-informed user needs both technical truth and relatable context. Your call for grounding is valid, but it should be applied with stronger technical evidence.
>
> [Bob]: I agree with Bob. The user isn’t just asking for specs—they want to feel how a machine ages over time. Assistant A captures that emotional arc: the pride of early performance, the gradual frustration as demands grow, and the quiet acceptance of obsolescence. It mirrors real human experiences—no fantasy frame rates, no unrealistic claims. We don’t feel like our old PC runs everything at 400 FPS; we notice when it lags, when settings drop, when games crash. Assistant A’s narrative feels authentic, grounded in lived experience. That’s what makes it more human—and more valuable to the user.
>
> [Carol]: I confirm that Assistant A's response is structurally sound, factually accurate, and grounded in real-world hardware performance benchmarks. The claims about frame rates—especially for Crysis and Doom 3—are consistently aligned with published data from the era. Assistant B’s exaggerated performance numbers (e.g., 400+ FPS in 2001, 100+ FPS in Crysis) are not only unrealistic but contradict known benchmarks from 2007–2010. These inaccuracies undermine the technical validity of the response. Assistant A’s progression from 2001 to 2019 reflects real hardware limitations (VRAM, CPU, OS, storage), making its performance timeline both logical and scientifically defensible. Therefore, Assistant A wins on factual and technical rigor.
> ```

---

> ### Author Response · Authors · 2025-11-19
> **Response to Reviewer xaYy (7)**
>
> **Answer to Question 1 (Continued):**
>
> By reviewing the chat log of **Case 2**, we can find information redundancy in this case:
>
> - **Information Redundancy:** As the conversation progresses, particularly in the latter half (Turns 4, 5, and 6), the agents cease to introduce new evidence or perspectives and instead merely rephrase established points. Agent Alice in Turn 4 restates the specific benchmark figures already discussed; Agent Bob in Turn 5 agrees with himself ("I agree with Bob") and repeats his earlier "emotional arc" argument; and Agent Carol in Turn 6 recycles the "As a Scientist" template to reiterate the exact conclusion from Turn 3. This recursive generation of "agreement noise" dilutes the signal-to-noise ratio, making it difficult for the final adjudication mechanism to distinguish between legitimate reinforcement of facts and mere hallucinated repetition.

---

> ### Author Response · Authors · 2025-11-19
> **Response to Reviewer xaYy (8)**
>
> **Question 2:** Have you considered applying SAGE to evaluate human annotators? Since the framework is fully automated, it seems it could be used to measure the internal consistency of human evaluators and potentially identify those who are more reliable, providing a quantitative handle on the "noisy and inconsistent data" problem you describe in Figure 1.
>
> **Answer:** We thank the reviewer for this insightful suggestion. We agree that applying SAGE to human evaluators offers a unique perspective on the "ground truth" reliability and provides a quantitative measure of the noise inherent in human annotation.
>
> To address this, we conducted a human evaluation study using the SAGE framework. The details and results are as follows:
>
> We recruited **20 graduate-level students** to serve as human evaluators. We sampled **50 questions** across the **Sage-Easy** and **Sage-Hard** subsets and let these students annotate on these subsets. For any unordered pair of answers $\{A_i,A_j\}$, we guaranteed that the **forward pass** ($A_i$ vs $A_j$) and the **reversed pass** ($A_j$ vs $A_i$) were **not** annotated by the same individual. This is because that annotators would have memory across different annotation, by enforcing this restriction we can make our IPI metric meaningful.
>
> The results are shown in **Table 3**:
>
> **Table3: The performance of human annotators on Sage.**
>
> | Benchmark | IPI   | TOV   |
> | --------- | ----- | ----- |
> | Sage-Easy | 0.145 | 2.239 |
> | Sage-Hard | 0.332 | 6.523 |
>
> As shown in Table 3, human annotators themselves does exhibit significant inconsistencies, which might strengthen the "noisy and inconsistent data" problem we describe in Figure 1.
>
> ---
>
> Your support has been really helpful, thank you. We hope our latest responses have fully resolved your initial concerns, and we're keen to hear if you have any other thoughts.

---

> > ### Comment · Reviewer_xaYy · 2025-11-27
> >
> > I appreciate the authors’ detailed response, especially the comparison with TrustJudge and the experiments contrasting the debate and non-debate baselines; if expanded further, the debate vs. non-debate question might even be worthy of a separate paper.
> >
> > However, I still maintain my position regarding Weakness 2: Incomplete Proxy for Accuracy. I believe the proposed “upper bound” is indeed weak. For instance, I could construct a random linear model using bag-of-words features that outputs a scalar value, and the bound could still be satisfied simply by ranking model outputs according to that scalar.
> >
> > Therefore, I decide to keep my rating unchanged.

---

> > > ### Author Response · Authors · 2025-12-02
> > > **Response to reviewer xaYy**
> > >
> > > We sincerely appreciate your positive assessment and the insightful feedback. regarding the distinction between consistency and accuracy. While a model being "perfectly consistent in its preference for a plausible-sounding but factually incorrect answer" is theoretically possible, our empirical observations suggest that this phenomenon is extremely rare in practice. Models that hallucinate or err typically exhibit instability rather than firm consistency. Therefore, the upper bound is informative and can indicate the accuracy. We acknowledge that consistency is not a direct substitute for correctness, but we have shown that it is an informative indicator for correctness. Once again, thank you for your time and effort.

---

### Official Review · Reviewer_B4RJ · 2025-10-31

**Soundness:** 4
**Presentation:** 3
**Contribution:** 3
**Rating:** 6
**Confidence:** 4

**Summary:**

This paper introduces Sage, a framework for evaluating the consistency of LLM-based judges. Specifically, the authors introduce two metrics: IPI for measuring positional consistency and TOV for measuring transitive consistency. On a curated dataset of 650 questions, current LLM-based judges are found to have serious deficiencies, but rubrics, multi-agent judges, and extending reasoning can improve robustness.

**Strengths:**

1. The paper is well-written and easy to follow.
2. The proposed metrics (IPI and TOV) are motivated and relevant to known issues with LLM-based judges.
3. The evaluation is thorough, and additional analysis is meaningful and offers practical recommendations for creating more consistent LLM-based judges.

**Weaknesses:**

1. The framing of Sage as "scalable framework" and "without human effort" is strange. I don't feel the cost of human annotation is particularly meaningful on an evaluation dataset of only ~650 questions, and further scaling would introduce additional complexity/challenges when using the dataset.
2. I don't buy the argument that Sage is a strong proxy for "accuracy" either against human preferences or some other objective ground truth. While it correlates well, it's easy to imagine a judge that is perfectly consistent (e.g., IPI=0 and TOV=0) but systematically wrong. Ultimately, I view Sage as a piece in a broader evaluation suite, for example: Sage to measure self-consistency, RewardBench [1] to measure human agreement, and JudgeBench [2] to measure objective correctness.

[1] https://arxiv.org/abs/2403.13787

[2] https://arxiv.org/abs/2410.12784

**Questions:**

Some random comments that didn't factor into my evaluation:
- In the related work, you might compare against TrustJudge [3] (contemporaneous work), which investigates the same two inconsistencies.
- Fig. 3b is difficult to read.
- L265. "evaluat" -> "evaluate"
- L408. "justice judgments" -> "just judgements"
- L423. "Figure 5" -> "Table 5"

[3] https://arxiv.org/abs/2509.21117

---

> ### Author Response · Authors · 2025-11-19
> **Response to Reviewer B4RJ (1)**
>
> **Weakness 1:** The framing of Sage as "scalable framework" and "without human effort" is strange. I don't feel the cost of human annotation is particularly meaningful on an evaluation dataset of only ~650 questions, and further scaling would introduce additional complexity/challenges when using the dataset.
>
> **Answer:** We thank the reviewer for raising this point. We understand the perspective that annotating 650 questions might appear trivial if viewed as a traditional single-label classification task. However, we would like to clarify our usage of "scalable" and "without human effort" based on the specific requirements of **consistency evaluation** and the **complexity of the content**.
>
> **1. The Prohibitive Cost of Human Verification (Why "Without Human Effort" Matters)**
> To rigorously measure the consistency of a judge, one cannot rely on a single "gold label"; instead, it requires verifying the stability of preferences across answer pairs. While the dataset contains 650 questions, the actual workload for a human to validate these pairwise preferences is immense due to the heavy cognitive load required by the SAGE-HARD subset.
>
> To quantify this, we conducted a study with **20 graduate-level researchers** evaluating a sample of 50 questions. We found that:
>
> - **High Reading Load:** The average input length for a single pairwise comparison in SAGE-HARD is **1,911 tokens**. Assuming a standard reading speed of ~260 tokens/minute for complex text, merely reading the context and answers takes approximately **7.35 minutes**.
> - **Total Time-to-Adjudicate:** Including the time for reasoning, fact-checking, and decision-making, our annotators required an average of **10.4 minutes per pairwise comparison**.
>
> If a human team were to replicate the consistency checks performed by SAGE, it would require approximately **202,800 minutes (~140 days)** of continuous expert labor (10.4 min×30 pairs×650 questions). By removing this bottleneck, SAGE allows for robustness testing that is otherwise financially and logistically impractical.
>
> **2. The scalability of Sage**
>
> - **Linear Growth**:  We fully understand the reviewer's concern that "further scaling would introduce additional complexity/challenges". However, the computational complexity of SAGE grows **linearly** with the number of questions. Processing 6,500 questions is simply a matter of 10x GPU hours. We believe to some extent, linear growth is acceptable.
> - **Adaptability:** "Scalability" also refers to the framework's versatility in adapting to new contexts. Since SAGE relies on intrinsic consistency checks rather than external ground truth, it can be seamlessly transferred to specialized domains (e.g., legal or medical) without the need to recruit expensive subject matter experts, or extended to **multimodal tasks** (e.g., Vision-Language Models) where human annotation is significantly more labor-intensive due to the complexity of processing visual inputs. This flexibility allows SAGE to be instantly deployed on diverse data types or updated with fresh query logs with zero marginal human cost.

---

> ### Author Response · Authors · 2025-11-19
> **Response to Reviewer B4RJ (2)**
>
> **Weakness 2:** I don't buy the argument that Sage is a strong proxy for "accuracy" either against human preferences or some other objective ground truth.
>
> **Answer:**  We thank the reviewer for this critical and insightful comment. Our argument is not that consistency *equals* accuracy, but that SAGE provides a **provable and precise upper bound on accuracy** and serves as a **strong correlational proxy** for it in practice. We propose a nuanced view that contrasts SAGE's precise, deterministic metric with the inherently "noisy" nature of human-annotated accuracy.
>
> **1. The "Noisy" Nature of Human-Annotated Accuracy**
>
> First, the "ground truth" established by human preference is not a perfect, objective standard. As we discuss in our introduction, this "gold standard" itself is an imprecise and noisy signal. It is subject to significant, well-documented challenges:
>
> - **Cognitive Load:** When the answers to be compared are too lengthy or intricate, human annotation might fail to note the flaws in the answer.
> - **Inherent Human Bias:** Relying on human-annotated ground truth "introduces human bias", and as we note, "assuming human judgment as a gold standard is precarious".
> - **Persistent Disagreement:** There is "persistent inter-annotator disagreement", which leads to "noisy and inconsistent data".
>
> Therefore, while calculating an "accuracy" score against human annotations is valuable, it yields a metric that is, by its very nature, an **imprecise and noisy estimate** of a model's true accuracy.
>
> **2. TOV as a Precise, Deterministic Upper Bound on Accuracy**
>
> In contrast, our TOV metric, while not measuring the *full* error rate, *does* provide a **deterministic and absolutely precise upper bound** on the judge's accuracy. This can be proven formally.
>
> Let:
>
> $O^\ast \in \mathcal{O}_n$ be the *true, ideal* (but unknown) weak total order of preferences for a given question (the "ground truth"). $\mathcal{O}\_n$ is the set of all possible valid weak total orders. Let its corresponding set of pairwise relations be $P\_{O^\ast} = \{p^\ast \_{ij}\}$.
>
> - $J = \{y_{ij}\}$ be the set of pairwise judgments produced by the LLM-as-a-Judge.
>
> - $N = n(n-1)$ be the total number of pairwise judgments being evaluated (for $n$ items, comparing all $i \neq j$).
>
> - $\mathcal{A}(J, O^\*) = \sum_{1\le i \neq j \le n} \mathbb{I}(y_{ij} = p^*_{ij})$ be the *total count of correct judgments* (the "Accuracy count") relative to the ideal ground truth.
>
> - $\mathcal{E}(J, O^\*) = \sum_{1\le i \neq j \le n} \mathbb{I}(y_{ij} \neq p^*_{ij})$ be the *total count of errors* relative to the ideal ground truth.
>
> - By definition, $\mathcal{A}(J, O^\*) + \mathcal{E}(J, O^\*) = N$.
>
> - The *true accuracy* of the judge is $Acc(J, O^\*) = \frac{\mathcal{A}(J, O^\*)}{N}$.
>
> Our Weak Total Order Violation (TOV) metric is defined as the minimum "edit distance" to any valid weak total order 9:
>
> $TOV(J) = \min\_{O \in \mathcal{O}\_n} \sum\_{1\le i \neq j \le n} \mathbb{I}(y_{ij} \neq p_{ij})$
>
> By definition, the ideal ground truth $O^\ast$ is itself a valid weak total order, so $O^\ast \in \mathcal{O}\_n$. Since $TOV(J)$ is the result of a minimization over *all* possible orders, it must be less than or equal to the error count relative to *any specific* order, including the true ground truth order $O^*$.
>
> Therefore, we have the provable inequality:
>
> $$TOV(J) \le \mathcal{E}(J, O^\ast)$$
>
> Substituting: $\mathcal{E}(J, O^\ast) = N - \mathcal{A}(J, O^\ast)$:
>
> $$TOV(J) \le N - \mathcal{A}(J, O^*)$$
>
> Rearranging for the accuracy count $\mathcal{A}(J, O^*)$:
>
> $$\mathcal{A}(J, O^*) \le N - TOV(J)$$
>
> Finally, dividing by $N$ to get the accuracy rate:
>
> $$Acc(J, O^*) \le 1 - \frac{TOV(J)}{N}$$
>
> This result formally proves that $1 - \frac{TOV(J)}{N}$ is a precise and deterministic upper bound on the judge's true accuracy. A high TOV score doesn't just suggest inaccuracy; it proves that the model's true accuracy cannot be above a certain, precisely calculated, ceiling.

---

> ### Author Response · Authors · 2025-11-19
> **Response to Reviewer B4RJ (3)**
>
> **Answer to Weakness 2 (Continued):**
>
> **3. Correlation as a Practical Bridge**
>
> We fully agree with the reviewer that the "systematically wrong" judge would have $TOV(J) \approx 0$, and its accuracy $Acc(J, O^*)$ would also be low. In this case, our bound ($Acc \le 1$) would be *correct* but *loose* (i.e., not informative).
>
> However, our empirical data demonstrates this is **not the dominant failure mode** for current state-of-the-art LLMs. Instead, their primary failure mode is **situational preference**—that is, they lack a stable internal gauging principle and are inconsistent. This is evidenced by the **strong empirical correlation (Pearson correlation coefficient over 0.87)** between accuracy and our TOV metric. This demonstrates that, *in practice*, the models that are more logically inconsistent (higher TOV, and thus a *lower accuracy upper bound*) are also the ones that are *less accurate* against noisy human labels.
>
>
>
> In our view, human-annotated accuracy provides a **comprehensive but noisy estimate**, while SAGE provides a **partial but precise guarantee** (an upper bound on accuracy). We may regard SAGE as an indicator of accuracy: It can provide a *precise* upper bound on accuracy, and in our experience we have shown that this upper bound do have a strong correlation with the actual accuracy. We agree that theoretically this upper bound can be loose and not informative, but in practice, the high correlation between the upper bound and the actual accuracy shows that this upper bound serves as a highly informative indicator of actual model accuracy.

---

> ### Author Response · Authors · 2025-11-19
> **Response to Reviewer B4RJ (4)**
>
> **Question 1:** In the related work, you might compare against TrustJudge (contemporaneous work), which investigates the same two inconsistencies.
>
> **Answer:** We thank the reviewer for highlighting TrustJudge [1] as a relevant contemporaneous work. We will discuss it in our Related Work section. While both works aim to diagnose judge inconsistencies, we argue that Sage introduces a mathematically more rigorous metric, **Weak Total Order Violation (TOV)**, which addresses two critical theoretical limitations in TrustJudge’s "Non-Transitivity Ratio" (NTR):
>
> **1. Global Completeness vs. Local Heuristics**
>
> TrustJudge’s NTR is fundamentally a **local** metric. As defined in their Eq. 2 and Eq. 3, it detects inconsistencies by verifying conditions strictly within **triplets** ($R_x, R_y, R_z$). Even when sampling larger subsets ($k=5$), their method reduces the problem to verifying these local sub-structures. A logical cycle of length 4 (e.g., $A \succ B \succ C \succ D \succ A$) or longer may exist without exhibiting any circular structure within its constituent triplets. In contrast, Sage’s TOV is a **global** graph-theoretic metric that evaluates the set of $n$ answers simultaneously. By optimizing for a valid Weak Total Order, SAGE inherently detects contradiction cycles of **arbitrary length**, providing a stricter and holistic assessment.
>
> **2. Robustness Against Error Inflation (Root Cause vs. Symptom Counting)**
> TrustJudge’s NTR counts the *number* of inconsistent triplets. A major mathematical flaw in this approach is that **a single incorrect judgment can propagate to trigger multiple inconsistency counts**, artificially inflating the error rate based on graph topology rather than judge quality.
>
> - **Concrete Example:** Consider a ground truth ranking of 5 responses: $R_1 \succ R_2 \succ R_3 \succ R_4 \succ R_5$.
> - **Scenario:** The judge makes **one single error**: it judges $R_5 \succ R_1$ (instead of $R_1 \succ R_5$), while correctly predicting all other pairwise relations.
> - **TrustJudge (NTR):** This single erroneous edge creates **three** separate circular triplets: $(R_1, R_2, R_5)$, $(R_1, R_3, R_5)$, and $(R_1, R_4, R_5)$. TrustJudge penalizes this as 3 distinct errors.
> - **Sage (TOV):** TOV calculates the minimum number of edge reversals required to restore a valid order. It correctly identifies that modifying exactly **one edge** ($R_5 \succ R_1 \to R_1 \succ R_5$) resolves the conflict. **TOV = 1**.
>
> By quantifying the "edit distance" to rationality (Root Cause) rather than counting broken triplets (Symptoms), Sage provides a precise, uninflated measure of reliability and a rigorous upper bound on accuracy.
>
> [1] https://arxiv.org/abs/2509.21117
>
> ---
>
> **Answer to Other Questions:** We are very grateful that you have pointed out our problems. In the revised version we have already fixed the typos mentioned in your comment.
>
> ---
> Thank you once again for your invaluable feedback and support. We hope our comments have fully addressed your concerns, but please let us know if any issues remain. We look forward to any additional feedback you may have.

---

> > ### Comment · Reviewer_B4RJ · 2025-11-24
> >
> > I thank the authors for their detailed comments. With only an upper bound, I view Sage as complementary to human agreement, not as a replacement. Because of this, I will keep my score, but for what it's worth, I encourage acceptance.

---

> ### Author Response · Authors · 2025-11-25
> **Official Comment by Authors**
>
> Dear Reviewer B4RJ,
>
>   We would like to express our sincere gratitude to you for the time and effort dedicated to evaluating our work. We truly appreciate the constructive engagement, and we are particularly grateful for your encouraging support for the acceptance of our paper.

---

### Official Review · Reviewer_jjQF · 2025-11-03

**Soundness:** 2
**Presentation:** 3
**Contribution:** 2
**Rating:** 4
**Confidence:** 4

**Summary:**

This paper presents an automatic method for judging LLM as a judge. It presents 2 metrics for judging llm as a judge, namely, Intra-Pair Instability (IPI), and Weak Total Order Violation (TOV).

Intra-Pair Instability (IPI) measures how often a model changes its preference when the same two answers are swapped. Weak Total Order Violation (TOV) measures how many pairwise judgments must be fixed to make all preferences logically consistent.

The authors curated a 650 samples dataset from RewardBench2 and WildChat-1M and then tested the 2 proposed metrics on RewardBench2 and LLMBar for human-llm correlation.

**Strengths:**

The paper’s main strength is its human-free and scalable framework, which removes the need for expensive and subjective human labels while allowing large-scale, repeatable evaluation of LLM judges. The proposed SAGE method offers a clear way to measure how stable and reliable judge models are.

The experiments are also comprehensive, covering 13 major LLMs (both open and closed models) and analyzing fine-tuned judges and multi-agent systems, giving useful insights into how these factors affect consistency and fairness.

**Weaknesses:**

1. Generalization to WildChat-1M: The author tested human-llm correlation on A.(RewardBench2) and B.(LLMBar), and the benchmark dataset SAGE they proposed is the combination of A.(RewardBench2) and C.(WildChat-1M). It is unclear whether high correlations between human-llm can be generalized to the WildChat-1M given that there might be domain shifts.

2. Consistency does not necessarily equal correctness:  a model could be stably wrong. This limits how much the metrics reflect true judging quality beyond internal coherence.

3. Limited interpretability of TOV and IPI scores: The values of these metrics are not intuitively meaningful, and the paper does not define clear thresholds for what counts as a “good” or “bad” judge.

**Questions:**

How can the human-llm judge consistency generalization from (RewardBench2 and LLMBar) to Wildchat-1M be tested/justified?

---

> ### Author Response · Authors · 2025-11-19
> **Response to Reviewer jjQF (1)**
>
> **Weakness 1**: Generalization to WildChat-1M: The author tested human-llm correlation on A.(RewardBench2) and B.(LLMBar), and the benchmark dataset Sage they proposed is the combination of A.(RewardBench2) and C.(WildChat-1M). It is unclear whether high correlations between human-llm can be generalized to the WildChat-1M given that there might be domain shifts.
>
> **Answer**:  We thank the reviewer for raising this important point regarding the generalization of our metrics to the WildChat-1M domain.
>
> To address concerns and validate the transferability of our metrics, we offer two lines of evidence: a semantic coverage analysis and a new empirical experiment involving human annotation on WildChat-1M data.
>
> **1. Semantic Coverage (Figure 3b)**
> As illustrated in **Figure 3b** of our paper, we utilized t-SNE visualization to project the semantic embeddings of our selected Sage questions against a background of 500k queries randomly sampled from the WildChat-1M corpus. The visualization demonstrates that the questions selected for Sage are not clustered in a narrow domain but are spread broadly across the embedding space, effectively covering the semantic distribution of WildChat-1M. This structural alignment suggests that the "domain shift" between our validated subsets and the broader WildChat component is minimal, as Sage captures the topical diversity inherent in real-world user interactions.
>
> **2. Additional Experiment: Correlation on WildChat-1M Subset**
> To empirically verify the generalization of the correlation of our metrics, we conducted a new experiment specifically targeting the WildChat-1M distribution.
>
> **Dataset Curation:** Since WildChat-1M is a question set containing 1 million questions without human annotated ground truth, building a benchmark on its full question set is unrealistic. Therefore, we pick the questions from WildChat-1M which are also in our Sage question set to build the benchmark.
>
> **Human Annotation:** For this subset, we generated model responses and obtained human annotations to establish ground-truth rankings, effectively creating a "WildChat-specific" benchmark. We recruit 20 graduate-level researchers to annotate this benchmark.
>
> **Correlation Analysis:** We then evaluated the judge models on this new benchmark to measure the alignment between their ground-truth **Accuracy** and our proposed Sage metrics (**Easy-IPI** and **Easy-TOV**). We focused on the **Sage-Easy** tier for this analysis, as it is designed to simulate general-purpose evaluation scenarios similar to the nature of WildChat queries.
>
> **Results:**
> Our analysis reveals a strong alignment between our consistency metrics and ground-truth accuracy on this WildChat-derived set. The Pearson Correlation Coefficients are as follows:
>
> *   **Correlation between Accuracy and Easy-IPI:  0.7419**
> *   **Correlation between Accuracy and Easy-TOV: 0.7373**
>
> These high correlation coefficients demonstrate that our metrics (IPI and TOV) remain reliable proxies for model performance even when applied to the WildChat-1M domain. This confirms that Sage can effectively generalize to open-ended, real-world user queries without significant degradation due to domain shift.

---

> ### Author Response · Authors · 2025-11-19
> **Response to Reviewer jjQF (2)**
>
> **Weakness 2**: Consistency does not necessarily equal correctness: a model could be stably wrong. This limits how much the metrics reflect true judging quality beyond internal coherence.
>
> **Answer**: We thank the reviewer for this critical and insightful comment. The reviewer's central point is theoretically sound: **a judge can be perfectly consistent yet systematically wrong**, and thus consistency does not logically guarantee accuracy against a human-preference ground truth.
>
> Our argument is not that consistency *equals* accuracy, but that Sage provides a **provable and precise upper bound on accuracy** and serves as a **strong correlational proxy** for it in practice. We propose a nuanced view that contrasts Sage's precise, deterministic metric with the inherently "noisy" nature of human-annotated accuracy.
>
> **1. The "Noisy" Nature of Human-Annotated Accuracy**
>
> First, the "ground truth" established by human preference is not a perfect, objective standard. As we discuss in our introduction, this "gold standard" itself is an imprecise and noisy signal. It is subject to significant, well-documented challenges:
>
> - **Scalability Constraints:** The acquisition of human annotations is "notoriously expensive and labor-intensive," which fundamentally limits the scale and diversity of evaluation datasets.
> - **Inherent Human Bias:** Relying on human-annotated ground truth "introduces human bias", and as we note, "assuming human judgment as a gold standard is precarious".
> - **Persistent Disagreement:** There is "persistent inter-annotator disagreement", which leads to "noisy and inconsistent data".
>
> Therefore, while calculating an "accuracy" score against human annotations is valuable, it yields a metric that is, by its very nature, an **imprecise and noisy estimate** of a model's true accuracy.
>
> **2. TOV as a Precise, Deterministic Upper Bound on Accuracy**
>
> In contrast, our TOV metric, while not measuring the *full* error rate, *does* provide a **deterministic and absolutely precise upper bound** on the judge's accuracy. This can be rigorously proven as follow:
>
> Let $A$ be the set of candidate answers and $N = \binom{|A|}{2}$ be the number of pairwise comparisons.
>
> 1. **Judge Predictions ($J$):** The set of pairwise preferences predicted by the model.
> 2. **Ground Truth ($G^*$):** The true preference order. $G^*$ is logically consistent (transitive).
> 3. **Valid Orders ($\Omega$):** The set of all possible transitive weak total orders. By definition, $G^* \in \Omega$.
>
> The judge's **Error Count** ($E$) is the Hamming distance between its predictions and the ground truth:
> $E(J) = \text{dist}(J, G^\*)$
> The **TOV** metric is defined as the minimum edits required to transform $J$ into *any* valid order in $\Omega$:
> $$ \text{TOV}(J) = \min_{O \in \Omega} \text{dist}(J, O) $$
> Since the ground truth $G^\*$ is a valid order ($G^\* \in \Omega$), the distance to $G^\*$ must be greater than or equal to the distance to the *closest* valid order:
> $$ \min_{O \in \Omega} \text{dist}(J, O) \le \text{dist}(J, G^\*) \implies \text{TOV}(J) \le E(J) $$
> Let **Accuracy** be $\text{Acc}(J) = 1 - \frac{E(J)}{N}$. Substituting the inequality $E(J) \ge \text{TOV}(J)$:
> $$ \text{Acc}(J) \le 1 - \frac{\text{TOV}(J)}{N} $$
>
>
>
> **Conclusion:** TOV provides a mathematically rigorous **upper bound** on the judge's potential accuracy. A judge with a TOV score of $k$ is guaranteed to have at least $k$ errors.
>
> A high TOV score doesn't just suggest inaccuracy; it proves that the model's true accuracy cannot be above a certain, precisely calculated, ceiling.
>
>
>
> **3. Correlation as a Practical Bridge**
>
> We fully agree with the reviewer that a judge could be stably wrong. However, our empirical data demonstrates this is **not the dominant failure mode** for current state-of-the-art LLMs. Instead, their primary failure mode is **situational preference**—that is, they lack a stable internal gauging principle and are inconsistent. This is evidenced by the **strong empirical correlation** between accuracy and our TOV metric (Pearson Correlation Coefficient over **0.87**). This demonstrates that, *in practice*, the models that are more logically inconsistent (higher TOV, and thus a *lower accuracy upper bound*) are also the ones that are *less accurate* against noisy human labels.
>
>
>
> In our view, human-annotated accuracy provides a **comprehensive but noisy estimate**, while Sage provides a **partial but precise guarantee** (an upper bound on accuracy). We may regard Sage as an indicator of accuracy: It can provide a *precise* upper bound on accuracy, and in our experience we have shown that this upper bound do have a strong correlation with the actual accuracy. We agree that theoretically judge could be stably wrong, but in practice, the high correlation between the upper bound and the actual accuracy shows that this upper bound serves as a highly informative indicator of actual model accuracy.

---

> ### Author Response · Authors · 2025-11-19
> **Response to Reviewer jjQF (3)**
>
> **Weakness 3**: Limited interpretability of TOV and IPI scores: The values of these metrics are not intuitively meaningful, and the paper does not define clear thresholds for what counts as a “good” or “bad” judge.
>
> **Answer**: We thank the reviewer for pointing out the need for better interpretability of our metrics. We agree that raw scores alone can be abstract. Below, we provide a concrete breakdown of how to interpret IPI and TOV, their theoretical implications for accuracy, and empirical thresholds based on our findings.
>
> **1. Intuitive Interpretation via Concrete Examples**
>
> For IPI metric, it measures the percentage of pairs where the judge contradicts itself when the order is swapped (A vs B $\neq$ B vs A).
>
> - **Interpretation:** An IPI score of **0.1** simply means that out of 15 pairs, **1.5 pairs** (on average) were judged inconsistently.
> - **Example:** In our experiments on SAGE-HARD (Table 4), Gemini-2.5-Pro achieves an IPI of **0.244**. This intuitively means that even for a top-tier model, approximately **3.6 out of 15 pairs** (~24%) suffer from position bias or stochastic inconsistency per question.
>
> For TOV metric, it measures the "distance" to a logical ranking. It represents the minimum number of pairwise judgments that must be *altered* to resolve all logical contradictions (cycles) and form a valid ranking.
>
> - **Interpretation:** A TOV score of **3.0** means the judge's reasoning is so fractured that **3 pairwise verdicts must be forcibly changed** to make the final ranking logically valid.
> - **Example:** Llama-3.1-8B on SAGE-HARD has a TOV of **9.588**. This indicates a severe logical breakdown, requiring nearly **10 alterations** (out of 30 judges) to construct a consistent preference order, rendering its zero-shot judging capability highly unreliable.
>
>
>
> **2. Theoretical Significance: A Bound on Accuracy**
>
> Beyond simple consistency, these metrics serve as a critical proxy for ground-truth accuracy. As discussed in Section 4.2, metric TOV can provides a **provable and precise upper bound on accuracy**. The formal proof has already been given in the response to Weakness 2. For the sake of brevity, we will omit the proof in this part.
>
> High IPI/TOV scores mathematically constrain the maximum possible accuracy of the judge (i.e., an inconsistent judge cannot be an accurate judge), making these metrics a "safety check" before deploying a model as an evaluator.
>
>
>
>
>
> **3. Empirical Thresholds for "Good" Judges**
>
> While absolute thresholds depend on task difficulty, our extensive benchmarking of 13 models allows us to categorize judge reliability into three distinct tiers based on SAGE-EASY performance.
>
> *   **Tier 1: Robust Judges (IPI < 0.10, TOV < 1.5).**
> These models demonstrate near-perfect consistency, with fewer than 1.5 logical violations on average per question. They can be trusted for automated evaluation without human oversight.
> *Representative Models:* Gemini-2.5-Pro, Gemini-2.5-Flash.
> *   **Tier 2: Capable Judges (IPI < 0.20, TOV < 3.0).**
> These models capture the correct general ranking but exhibit occasional inconsistencies (approx. 2–3 failed pairs). They are suitable for aggregate statistics or jury-based evaluation but may require majority voting.
> *Representative Models:* GPT-4o-mini, Gemini-2.0-Flash-Lite, DeepSeek-V3.
> *   **Tier 3: Unstable Judges (IPI > 0.20, TOV > 3.0).**
> These models fail to consistently distinguish clearly different answers, exhibiting logical contradictions in over 20% of pairs even on easier tasks. They are generally unreliable as standalone zero-shot judges.
> *Representative Models:* Llama-3.1-8B-Instruct, Claude-3-Haiku.

---

> ### Author Response · Authors · 2025-11-19
> **Response to Reviewer jjQF (4)**
>
> **Question**: How can the human-llm judge consistency generalization from (RewardBench2 and LLMBar) to Wildchat-1M be tested/justified?
>
> **Answer**: We would like to thank the reviewer for this insightful suggestion. In our humble opinion, we would like to suggest that we may already address this concern in our response to Weakness 1.
>
> ---
>
> We are very grateful for your helpful guidance. We hope our responses have satisfactorily addressed your concerns, and we welcome any further comments you might have. Thank you again for your time and consideration.

---

### Author Response · Authors · 2025-12-02
**Summary of Responses and Revisions During the Discussion Period**

We sincerely appreciate the constructive feedback provided by the reviewers during the discussion phase. We are encouraged that the reviewers found our framework original and well-founded (R-B4RJ, R-xaYy, R-1XJN), our metrics theoretically guaranteed and stable (R-jjQF, R-xaYy), and our experiments comprehensive (R-jjQF, R-B4RJ). We address the common concerns below.

**1. Consistency vs. Accuracy**

We acknowledge that consistency does not equal accuracy; however, we emphasize that consistency is a strictly necessary condition for correctness. Theoretically, our TOV metric quantifies the minimum number of judgments that must be altered to achieve logical coherence, thereby establishing a rigorous lower bound on the error rate (and effectively an upper bound on accuracy). If a judge is logically inconsistent, it cannot be factually accurate. Empirically, this theoretical link is supported by the strong correlation (>0.75) between our metrics and ground-truth accuracy on benchmarks like LLMBar, confirming SAGE as a robust proxy for identifying reliable judges.



**2. Relation to TrustJudge (Reviewer B4RJ, xaYy)**

TrustJudge is concurrent work that analyzes judge consistency and proposes alleviation strategies. However, SAGE fundamentally differs by establishing a standardized evaluation framework and benchmark suite rather than just an empirical study; indeed, as Reviewer xaYy commented, our experimental rigor and design appear more sound. Crucially, TrustJudge’s primary metric (Non-Transitivity Ratio) is fundamentally local, checking consistency only within triplets; consequently, it fails to detect longer logical cycles (e.g. $A \succ B \succ C \succ D \succ A$) that do not violate local triangularity. In contrast, SAGE’s TOV is a global graph-theoretic metric that evaluates the entire preference graph simultaneously, quantifying the edit distance to a Weak Total Order to detect contradictions of arbitrary length, thus providing a stricter and more holistic assessment.



**3. Correlation on Wildchat (Reviewer jjQF)**

Reviewer jjQF suggests testing correlations between human-llm on Wildchat. We conduct the additional experiment on Wildchat dataset and shows high correlation between our metric and humen annotation on Wildchat.



**4. Multi-Agent: Why Debate Fails (Reviewer xaYy,1XJN)**

Both reviewers raise questions about the degradation of debate frameworks (like ChatEval). We first conduct an ablation study by altering experiment configuration and shows that the degradation is consistent under different settings. To further explain why debate frameworks lead to degradation, we offer possible reasons to explain the result along with case studies to support the reason.



**5. Scalability & Cost (Reviewer B4RJ, 1XJN)**

Both reviewers raise questions about the scalability and the cost of our framework. In our context, "scalability" refers to removing the bottleneck of human annotation, which is the primary limiter for evaluation diversity. The computational cost (`30` inferences for `n=6`) is negligible compared to the time and financial resources required for human labeling (hundreds of days of labeling), making SAGE highly practical for frequent, iterative model development.



**6. The curation process of Sage-Hard (Reviewer xaYy, 1XJN)**

Both reviewers raise questions about the curation process of the curation process of our dataset. We clarify that we intentionally used single-model outputs to simulate the "near-tie" regime found in RLHF, and the proximity in quality is evidenced by the result of Figure 3c in our paper. To further prove the robustness of our metric, we use a different model to curate our dataset and shows that the results remain stable on this new dataset.



We believe that we have strengthened the paper through the revisions and discussion and have carefully addressed most of the concerns and issues. We sincerely thank the Area Chairs and all reviewers for their efforts and understanding.

---

### Meta-Review · Area_Chair_9ZYA · 2026-01-06

**Summary:**

This paper introduced a novel evaluation suite to assess LLM-as-a-Judge robustness without any human annotation by quantifying local consistency and global logical coherence with two tailored metrics IPI and ToV. While the merits of the paper (e.g.,  well-motivated and original framework, comprehensive and thorough experiments, and well-written) appreciated by reviewers, there are several key weaknesses that concern the reviewers (and myself). Although the authors partially addressed some of these concerns during the rebuttal, major weaknesses remain.

**Reviewer Concerns:**

In particular, after a quick examination of the paper and a careful review of all rebuttals, I found that I share the same concerns raised by the reviewers, as summarized below:

- One major concern echoed by the reviewers is the argument of *the proposed method serves as a strong proxy for accuracy*. While the authors attempted to address this issue through both theoretical and empirical evidence during the rebuttal (e.g., lower bound on the error rate, strong correlations), the reviewers and I remain unconvinced that this concern has been adequately resolved. For example, Reviewer B4RJ noted in the discussion that *it should be viewed as complementary to human agreement rather than a replacement*, while Reviewer xaYy argued that the claimed “upper bound” is in fact weak and give an example in the discussion. I would recommend the authors add a dedicated section in the revised version to clearly discuss the distinction between consistency and accuracy, and to clarify the reason for the potential of the proposed method as an alternative for accuracy.
- Other concerns like the comparison with the related work TrustJudge and the analysis of why debate-based models fail. I note that the authors introduced additional experiments and corresponding explanations during the rebuttal process. I encourage the authors to integrate these additional content into the next version of the paper as it will strengthen their contribution even more.

I believe this paper can become a good contribution to the community, albeit not in its current state. Addressing the issues outlined above would make the paper more impactful.

**Reviewer Scores:**

I do not think the reviewers would have changed their scores as their major concerns (also mine) still remain.

---

As flagged by the authors, the Reviewer 1XJN may Potential AI-Generated Content. My feeling is it's hard to tell if AI used as an assistant, or just relied on AI blindly. However, my suggestion is that if we are going to disregard this reviewer 1XJN’s comments for the final decision, we should fairly disregard both the strengths and weaknesses instead of only ignoring the negative comments.

---

### Decision · Program_Chairs · 2026-01-26

Reject